# Pretraining task diversity and the emergence of non-Bayesian in-context learning for regression

**Allan Raventós**[*]    **Mansheej Paul**[*]    **Feng Chen**    **Surya Ganguli**
Stanford University
{aravento, mansheej, fengc, sganguli}@stanford.edu

## Abstract

Pretrained transformers exhibit the remarkable ability of in-context learning (ICL): they can learn tasks from just a few examples provided in the prompt without updating any weights. This raises a foundational question: can ICL solve fundamentally *new* tasks that are very different from those seen during pretraining? To probe this question, we examine ICL's performance on linear regression while varying the diversity of tasks in the pretraining dataset. We empirically demonstrate a *task diversity threshold* for the emergence of ICL. Below this threshold, the pretrained transformer cannot solve unseen regression tasks, instead behaving like a Bayesian estimator with the *non-diverse pretraining task distribution* as the prior. Beyond this threshold, the transformer significantly outperforms this estimator; its behavior aligns with that of ridge regression, corresponding to a Gaussian prior over *all tasks*, including those not seen during pretraining. Thus, when pretrained on data with task diversity greater than the threshold, transformers *can* optimally solve fundamentally new tasks in-context. Importantly, this capability hinges on it deviating from the Bayes optimal estimator with the pretraining distribution as the prior. This study also explores the effect of regularization, model capacity and task structure and underscores, in a concrete example, the critical role of task diversity, alongside data and model scale, in the emergence of ICL.

## 1 Introduction

Pretrained transformers (PTs) can learn new tasks from just a few examples provided in the prompt without taking any gradient steps on those examples [1]. This ability, called *in-context learning (ICL)*, has unlocked the widspread use of language models by making it efficient to adapt general purpose models to bespoke tasks without explicit training. Though remarkable, what makes ICL mysterious, and potentially harmful [2], is that the learning algorithm implemented by the PT in its forward pass is not built into its architecture or training process; instead it emerges from pretraining on large-scale data with a next token prediction objective. This raises a foundational question: can ICL really solve fundamentally *new* tasks that are very different from those seen during pretraining? If so, what learning algorithm does ICL implement? To answer these questions, we need to better understand how the different ingredients that go into pretraining influence this ability.

Towards this end, we explore how the diversity of tasks in the pretraining data affects the emergence of ICL. Prior work [3] has proposed that ICL works by performing Bayesian inference. During pretraining, transformers learn a prior over latent tasks represented in the pretraining data. When prompted with examples at inference time, they "retrieve" relevant pretraining tasks and generate subsequent tokens from the posterior distribution conditioned on the query and inferred tasks. This suggests that ICL performance on a new task is influenced by its similarity to tasks implicitly learned during pretraining. However, the distribution of tasks in our pretraining data, $\mathcal{T}_{\text{Pretrain}}$, is usually a

---

[*]Equal Contribution. Code released at `https://github.com/mansheej/icl-task-diversity`

37th Conference on Neural Information Processing Systems (NeurIPS 2023).

limited and unrepresentative subsample of the ideal distribution of tasks, $\mathcal{T}_{\text{True}}$, that we want our model to be capable of learning in-context. For instance, $\mathcal{T}_{\text{True}}$ could be the set of all instructions we want an A.I. assistant to follow. But, large-scale language modeling datasets [4, 5] used to pretrain these models contain very few examples correctly formatted for ICL. Instruction finetuning (IFT) datasets [6–11] designed to ameliorate this are expensive to collect and thus contain tasks from just a few domains. Under the Bayesian framework, this distribution mismatch would cause the Bayesian estimator with a prior over the limited pretraining tasks, $\mathcal{T}_{\text{Pretrain}}$, to perform suboptimally on tasks that are very different from those seen during pretraining. This motivates our question: can a model pretrained on a dataset with low task diversity nevertheless learn *new, unseen* tasks?

For general purpose language modeling, the size and complexity of $\mathcal{T}_{\text{Pretrain}}$ and the vague specification of $\mathcal{T}_{\text{True}}$ make this question challenging to analyze. So, following recent work [12–14], we study ICL for linear regression. Here, a task is a linear regression problem with a given latent regression vector; the PT must predict the target for a new data point from examples of data-target pairs provided in the prompt. Prior work [13] has shown that transformers that see an *unlimited* number of latent regression vectors during pretraining learn to perform ridge regression with the Bayes optimal ridge parameter. We instead consider the case where the pretraining task distribution, $\mathcal{T}_{\text{Pretrain}}$, contains a *limited and finite* set of latent regression vectors (see Section 2 for details). To evaluate its ability to learn new tasks, the PT is tested on the ideal task distribution, $\mathcal{T}_{\text{True}}$, which is a Gaussian distribution over *all* latent regression vectors. Studying this setting has two advantages: first, we can directly vary the task diversity in $\mathcal{T}_{\text{Pretrain}}$ by changing the number of unique latent regression vectors seen during pretraining. Second, we can calculate the optimal estimator that minimizes the pretraining loss—the Bayesian estimator with prior $\mathcal{T}_{\text{Pretrain}}$—as well as the optimal estimator for all tasks—the Bayesian estimator with prior $\mathcal{T}_{\text{True}}$. This allows us to interpret the behavior of the PT by comparing its predictions to those of the optimal estimators under either task distribution. In our work, we vary the pretraining task diversity and probe the PT's ability to learn fundamentally new tasks in-context: does it behave like the optimal estimator for $\mathcal{T}_{\text{Pretrain}}$ and perform suboptimally on tasks from $\mathcal{T}_{\text{True}}$, or does it align with the optimal estimator for $\mathcal{T}_{\text{True}}$ which can solve new, unseen tasks?

**Contributions.** Our contributions are as follows:

- We find that a transformer pretrained on data with low task diversity behaves like the Bayesian estimator with prior $\mathcal{T}_{\text{Pretrain}}$; it performs optimally on pretraining tasks but cannot learn new tasks in-context. However, as pretraining task diversity increases, the PT deviates from this Bayesian estimator, significantly outperforming it on new tasks, and at a large but still finite number of pretraining tasks, the PT's performance closely matches that of the optimal estimator on $\mathcal{T}_{\text{True}}$.

- We identify a task diversity threshold for the emergence of ICL. Below this threshold, increasing the pretraining dataset size while keeping task diversity constant biases the PT towards the pretraining task distribution. Conversely, beyond this threshold, increasing the dataset size without increasing its task diversity improves the PT's performance on new, unseen tasks. This suggests that the PT's behavior undergoes a sharp algorithmic phase transition in the limit of many examples per task, aligning with the optimal estimators on $\mathcal{T}_{\text{Pretrain}}$ before the threshold and on $\mathcal{T}_{\text{True}}$ after it. We also examine this transition from the perspective of learning dynamics.

- We empirically show that increasing the task dimension at fixed SNR increases the task diversity threshold. However, the scaling of the PT's error with dimension is vastly superior to that of the optimal Bayesian estimator for $\mathcal{T}_{\text{Pretrain}}$; at a task diversity that is beyond the threshold at all dimensions we consider, the PT remains near-optimal with increasing dimension, whereas the optimal estimator for $\mathcal{T}_{\text{Pretrain}}$ grows progressively less similar to the optimal estimator for $\mathcal{T}_{\text{True}}$.

- We show that increasing weight decay significantly decreases the task diversity threshold while increasing number of layers or embedding size increases the task diversity threshold. This elucidates the effect of regularization and model capacity on the emergence of ICL.

Overall these contributions suggest that the emergence of in-context learning in pretrained transformers cannot be fully explained by a theory of Bayesian inference on the pretraining distribution.

## 2    Problem setup

**ICL of linear regression (schematic in Fig. 1).** Each ICL *task* corresponds to a latent $D$-dimensional regression vector, $\mathbf{w} \in \mathbb{R}^D$. At inference time, the transformer takes as input a sequence of $K$ data-target pairs, $(\mathbf{x}_1, y_1, ..., \mathbf{x}_K, y_K)$, which are the in-context examples corresponding to this *single* task

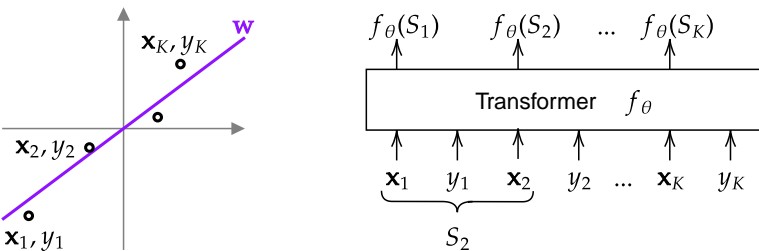

Figure 1: **Schematic for ICL of linear regression.** *(Left)* A task corresponds to a latent regression vector, $\mathbf{w}$ (purple line). $(\mathbf{x}_1, y_1, ..., \mathbf{x}_K, y_K)$ (black circles) is a sequence of in-context examples for this task. *(Right)* The PT, $f_\theta$, takes this as input and generates $K$ outputs. The $k$th output, $f_\theta(S_k)$, is the prediction for the target of $\mathbf{x}_k$ and depends only on the context $S_k = (\mathbf{x}_1, y_1, ..., \mathbf{x}_{k-1}, y_{k-1}, \mathbf{x}_k)$.

$\mathbf{w}$. For $k \in \{1, ..., K\}$, the data are drawn i.i.d. from a $D$-dimensional standard normal distribution, $\mathbf{x}_k \sim \mathcal{N}(\mathbf{0}, \mathbf{I}_D)$, and the targets are scalars given by $y_k = \mathbf{w}^\intercal \mathbf{x}_k + \varepsilon_k$. The $\varepsilon_k$'s are noise scalars drawn i.i.d. from a normal distribution with mean 0 and variance $\sigma^2$, $\varepsilon_k \sim \mathcal{N}(0, \sigma^2)$. Let $f_\theta$ be the PT with parameters $\theta$. Since we use a decoder-only transformer with a causal attention mask, for each $k \in \{1, ..., K\}$, the transformer sees the context $S_k = (\mathbf{x}_1, y_1, ..., \mathbf{x}_{k-1}, y_{k-1}, \mathbf{x}_k)$ and based on this context, it makes a prediction, $f_\theta(S_k)$, for the target of $\mathbf{x}_k$. Thus, in each forward pass, the PT solves $K$ linear regression problems each with the same latent regression vector but an increasing number of in-context examples.

**Pretraining.** The transformer is pretrained to minimize the next token prediction mean squared error (MSE) on sequences of data and target pairs. The latent regression vector for each sequence is drawn from the **pretraining task distribution**, $\mathcal{T}_{\text{Pretrain}}$. This distribution has limited diversity as it is the uniform distribution over a *finite* set of $M$ tasks, $\mathcal{T}_{\text{Pretrain}} = \mathcal{U}\{\mathbf{w}^{(1)}, ..., \mathbf{w}^{(M)}\}$. Each task in $\mathcal{T}_{\text{Pretrain}}$ is drawn i.i.d from a $D$-dimensional standard normal distribution, $\mathbf{w}^{(i)} \sim \mathcal{N}(\mathbf{0}, \mathbf{I}_D)$, $i \in 1, ..., M$. By increasing the number of tasks, $M$, in $\mathcal{T}_{\text{Pretrain}}$, we can increase the diversity of the pretraining data. Since the transformer makes a prediction for every data point in the sequence, its loss, $L^{\mathcal{T}_{\text{Pretrain}}}$, is just the MSE for each prediction, averaged over the predictions in the sequence:

$$L^{\mathcal{T}_{\text{Pretrain}}}(\theta) = \underset{\substack{\mathbf{w} \sim \mathcal{T}_{\text{Pretrain}} \\ \mathbf{x}_1, ..., \mathbf{x}_K \sim \mathcal{N}(\mathbf{0}, \mathbf{I}_D) \\ \varepsilon_1, ..., \varepsilon_K \sim \mathcal{N}(0, \sigma^2)}}{\mathbb{E}} \left[ \frac{1}{K} \sum_{k=1}^{K} (f_\theta(S_k) - y_k)^2 \right]. \tag{1}$$

**Evaluation.** We evaluate the PT's performance on *tasks seen during pretraining* by computing $L^{\mathcal{T}_{\text{Pretrain}}}$ using Eq. (1) but with new samples of data and noise. Since these are new instances of the task with new in-context examples, this evaluation corresponds to the test error of the PT. For a PT to successfully perform ICL of linear regression on *new tasks*, it must accurately predict the targets from the in-context examples for any task drawn from an **ideal task distribution**, $\mathcal{T}_{\text{True}}$, over all latent regression vectors; in our case $\mathcal{T}_{\text{True}} = \mathcal{N}(\mathbf{0}, \mathbf{I}_D)$. We evaluate the PT's performance on new tasks by computing $L^{\mathcal{T}_{\text{True}}}$, which follows Eq. (1) but where the tasks are sampled from the ideal task distribution: $\mathbf{w} \sim \mathcal{T}_{\text{True}}$ in the expectation.

**Comparing the PT to optimal estimators.** An advantage of studying ICL of linear regression is that we can calculate the ground truth optimal estimators that minimize the loss, $L^{\mathcal{T}}$, in Eq. (1) for both task distributions, $\mathcal{T}_{\text{Pretrain}}$ and $\mathcal{T}_{\text{True}}$. The optimal estimator for the $k$th prediction, $\hat{y}_k^{\mathcal{T}}$, that minimizes the $k$th term in the sum in $L^{\mathcal{T}}$ is the Bayesian estimator with $\mathcal{T}$ as the prior. This is given by the posterior mean of $y_k$ conditioned on the context: $\hat{y}_k^{\mathcal{T}} = \mathbb{E}_{\mathcal{T}, \varepsilon_k}[y_k \mid S_k]$, where the expectation is over the task distribution, $\mathcal{T}$, and the noise, $\varepsilon_k$ (Appendix A.1).

For task distribution $\mathcal{T}_{\text{Pretrain}} = \mathcal{U}\{\mathbf{w}^{(1)}, ..., \mathbf{w}^{(M)}\}$, the discrete minimum mean squared error (**dMMSE**) estimator is optimal. It is given by $\hat{y}_k^{\text{dMMSE}} = (\hat{\mathbf{w}}_k^{\text{dMMSE}})^\intercal \mathbf{x}_k$ where $\hat{\mathbf{w}}_1^{\text{dMMSE}} =$

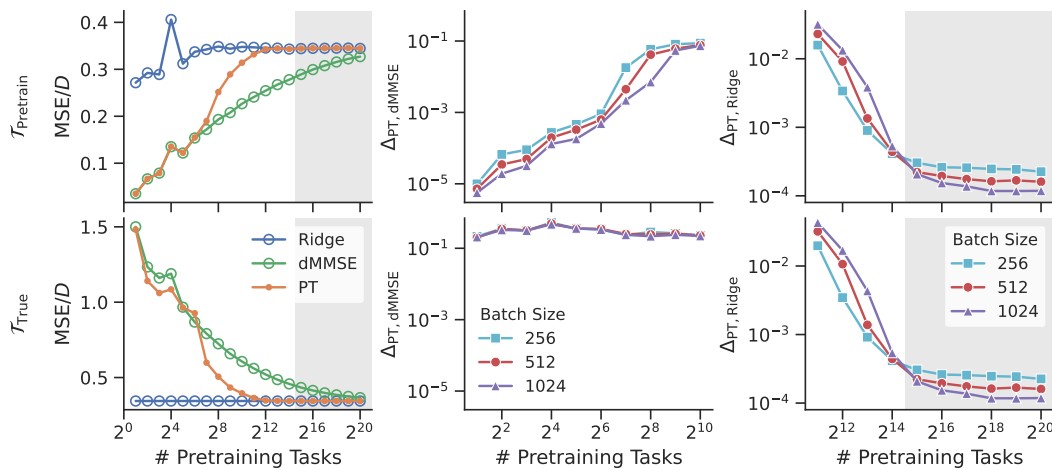

Figure 2: **ICL emerges in PTs beyond a threshold pretraining task diversity.** We show all results on both tasks seen during pretraining (*top row*) and on new tasks (*bottom row*). The *left column* compares the normalized loss of transformers pretrained with increasing task diversity to that of dMMSE and Ridge. When the pretraining task diversity is small, the PT's performance matches that of dMMSE; it performs very well on tasks seen during pretraining but poorly on new tasks. As the pretraining task diversity increases, both dMMSE and PT approach Ridge. However, the PT approaches Ridge much faster, significantly outperforming dMMSE on new tasks *(bottom left)*. In the *middle and right columns*, we compare the PT's predictions to those of dMMSE and Ridge respectively (Eq. (4)). We also increase the number of sequences per task at each level of task diversity by increasing the batch size while keeping total training steps fixed. This reveals a task diversity threshold between $2^{14}$ and $2^{15}$ pretraining tasks at which there is a phase transition in the behavior of the model. Below the threshold, increasing the dataset size leads to PTs with predictions more aligned with dMMSE on $\mathcal{T}_{\text{Pretrain}}$ *(top middle)*. However, beyond this threshold (indicated by gray shading), increasing the dataset size leads to PTs more aligned with Ridge on all tasks *(right)*.

$\frac{1}{M} \sum_{i=1}^{M} \mathbf{w}^{(i)}$ and for $k \in \{2, ..., K\}$, (Appendix A.2)

$$\hat{\mathbf{w}}_k^{\text{dMMSE}} = \sum_{i=1}^{M} \frac{\exp\left(-\frac{1}{2\sigma^2} \sum_{j=1}^{k-1} (y_j - \mathbf{w}^{(i)\intercal}\mathbf{x}_j)^2\right)}{\sum_{l=1}^{M} \exp\left(-\frac{1}{2\sigma^2} \sum_{j=1}^{k-1} (y_j - \mathbf{w}^{(l)\intercal}\mathbf{x}_j)^2\right)} \mathbf{w}^{(i)}. \tag{2}$$

Intuitively, $\hat{\mathbf{w}}_k^{\text{dMMSE}}$ is just a weighted sum of the pretraining $\mathbf{w}^{(i)}$s with weight governed by the likelihood of observing targets $\{y_1, ..., y_{k-1}\}$ conditioned on inputs $\{\mathbf{x}_1, ..., \mathbf{x}_{k-1}\}$ and the task being $\mathbf{w}^{(i)}$. A PT that minimizes the pretraining loss $L^{\mathcal{T}_{\text{Pretrain}}}$ will behave like this estimator.

For task distribution $\mathcal{T}_{\text{True}} = \mathcal{N}(\mathbf{0}, \mathbf{I}_D)$, the **Ridge** regression estimator with the ridge parameter set to the noise scale $\sigma^2$ is optimal: $\hat{y}_k^{\text{Ridge}} = \left(\hat{\mathbf{w}}_k^{\text{Ridge}}\right)^\intercal \mathbf{x}_k$, where $\hat{\mathbf{w}}_1^{\text{Ridge}} = \mathbf{0}$ and for $k = \{2, ..., K\}$,

$$\hat{\mathbf{w}}_k^{\text{Ridge}} = \left(\mathbf{X}^\intercal\mathbf{X} + \sigma^2\mathbf{I}_D\right)^{-1} \mathbf{X}^\intercal\mathbf{y}, \tag{3}$$

where $\mathbf{X} = (\mathbf{x}_1^\intercal, ..., \mathbf{x}_{k-1}^\intercal) \in \mathbb{R}^{(k-1)\times D}$ and $\mathbf{y} = (y_1, ..., y_{k-1})$ (Appendix A.3). A PT that performs optimally on new tasks will behave like this estimator. We can compare the behavior of the PT to that of the optimal estimators by computing the mean square difference of the predictions under a given task distribution $\mathcal{T}$. We write this as

$$\Delta_{\text{PT,Ridge/dMMSE}}^{\mathcal{T}} = \mathop{\mathbb{E}}_{\substack{\mathbf{w}\sim\mathcal{T} \\ \mathbf{x}_1,...,\mathbf{x}_K\sim\mathcal{N}(\mathbf{0},\mathbf{I}_D) \\ \varepsilon_1,...,\varepsilon_K\sim\mathcal{N}(0,\sigma^2)}} \left[\frac{1}{KD} \sum_{k=1}^{K} \left(f_\theta(S_k) - \hat{y}_k^{\text{Ridge/dMMSE}}\right)^2\right]. \tag{4}$$

## 3 Experiments and results

Unless specified otherwise, we study linear regression in $D = 8$ dimensions with up to $K = 16$ in-context examples and observation noise variance $\sigma^2 = 0.25$. We use either a *base* transformer

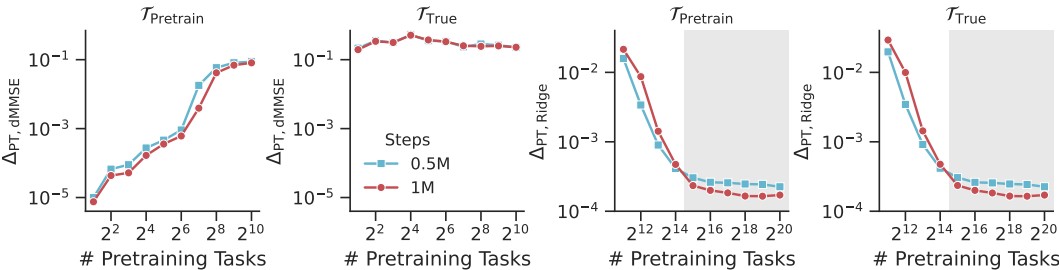

Figure 3: **Increased pretraining steps reveals the same task diversity threshold for the emergence of ICL.** *Columns 1 and 2* in this figure are similar to the middle column in Fig. 2 and *columns 3 and 4* correspond to the right column in Fig. 2, except here we increase the number of sequences per task by increasing the number of training steps while keeping batch size = 256. Both methods of increasing dataset size—increasing batch size in Fig. 2 and increasing training steps in this figure—reveal a transition in the behavior of the PT: beyond the task diversity threshold, ICL on new tasks emerges.

model with the GPT2 architecture [15] with 8 layers, 128-dimensional embeddings, and 2 attention heads or a *small* model with 4 layers, 64-dimensional embeddings, and 2 attention heads. We train with the Adam optimizer [16] and a one-cycle triangle learning rate schedule [17] with 50% warmup. The *base* model is trained with batch size 256 for 500K training steps, though these hyperparameters are varied in our experiments. We always sweep over a range of learning rates and choose the largest learning rate at which training is stable. For further details see Appendix B.

To pretrain a randomly initialized transformer on data with task diversity $M$, we first construct the pretraining task distribution, $\mathcal{T}_{\text{Pretrain}}$, as described in Section 2. We then minimize the objective $L^{\mathcal{T}_{\text{Pretrain}}}$ in Eq. (1) using minibatch stochastic gradient descent. For each sequence in a minibatch, we sample a single task $\mathbf{w}$ from $\mathcal{T}_{\text{Pretrain}}$, as well as *new* samples of data, $\{\mathbf{x}_i\}_{i=1}^{K}$, and noise, $\{\varepsilon_i\}_{i=1}^{K}$, from their respective continuous distributions, to form a sequence $(\mathbf{x}_1, \mathbf{w}^\intercal \mathbf{x}_1 + \varepsilon_1, \ldots, \mathbf{x}_K, \mathbf{w}^\intercal \mathbf{x}_K + \varepsilon_K)$. If we train for $N$ steps at batch size $B$, the transformer will see a total of $NB$ *unique* sequences and roughly $\frac{NB}{M}$ unique sequences for each latent task in $\mathcal{T}_{\text{Pretrain}}$. By increasing either $B$ or $N$ at fixed $M$, we can increase the total size of the pretraining dataset (or number of sequences per task) while keeping the dataset *diversity*—the number of unique $\mathbf{w}$s in $\mathcal{T}_{\text{Pretrain}}$—fixed.

### 3.1 Task diversity threshold for the emergence of in-context learning

For Fig. 2, we pretrain our base transformer on datasets with increasing task diversity (on the x-axis) while keeping the total number of sequences seen during pretraining fixed ($B = 256, N = 500K$). We evaluate the PTs and both optimal estimators on tasks seen during pretraining drawn from $\mathcal{T}_{\text{Pretrain}}$ (Fig. 2 top left) and on new tasks drawn from $\mathcal{T}_{\text{True}}$ (Fig. 2 bottom left) and plot MSE normalized by task dimension—$L^{\mathcal{T}}/D$ from Eq. (1)). Since dMMSE is optimal on tasks from $\mathcal{T}_{\text{Pretrain}}$ (as discussed in Section 2), the green dMMSE markers denote the lowest possible loss the PT could achieve in this setting. In fact, the pretraining objective $L^{\mathcal{T}_{\text{Pretrain}}}$ *explicitly encourages* the PT to match dMMSE performance. On the other hand, Ridge is optimal on tasks sampled from $\mathcal{T}_{\text{True}}$ (Fig. 2 bottom left); the blue markers denote the lowest possible MSE the PT could attain on *new* tasks.

**Low task diversity phase: the PT is Bayesian with respect to the pretraining distribution and cannot solve new tasks.** At low pretraining task diversity—$M$ up to about $2^6$—the PT's MSE closely tracks that of dMMSE on tasks sampled from $\mathcal{T}_{\text{Pretrain}}$ (Fig. 2 top left); the PT performs optimally on tasks seen during pretraining. But it significantly *underperforms* on new tasks sampled from $\mathcal{T}_{\text{True}}$, indicated by the gap in MSE between the PT and Ridge (Fig. 2, bottom left). In this regime, it behaves like the Bayesian estimator with prior $\mathcal{T}_{\text{Pretrain}}$.

**High task diversity phase: the PT is non-Bayesian with respect to the pretraining task distribution and can solve new tasks.** At higher task diversities—above $2^6$ pretraining tasks—the PT's MSE deviates from dMMSE and approaches Ridge under *both* $\mathcal{T}_{\text{Pretrain}}$ and $\mathcal{T}_{\text{True}}$. Crucially, the PT starts to significantly *outperform* dMMSE on unseen tasks sampled from $\mathcal{T}_{\text{True}}$ (Fig. 2 bottom left) at the expense of not fully minimizing its training objective, $L^{\mathcal{T}_{\text{Pretrain}}}$ (gap between PT and dMMSE under $\mathcal{T}_{\text{Pretrain}}$, Fig. 2 top left). This suggests that, a PT trained on a finite but large number of pretraining tasks can learn fundamentally new tasks in-context and this ability depends on it deviating from the optimal Bayesian estimator on the pretraining task distribution.

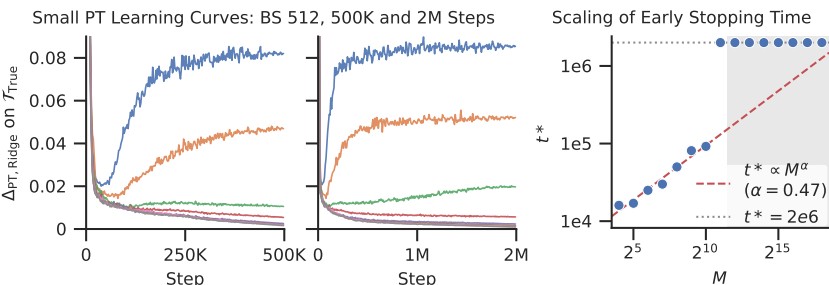

Figure 4: **Learning dynamics of *small* PTs shows a transition at the task diversity threshold.** We plot $\Delta_{\text{PT,Ridge}}^{\mathcal{T}_{\text{Pretrain}}}$ vs training steps for *small* PTs. For the same $M$, learning curves for short (500K steps, **left**) or long (2M steps, **center**) training durations are similar, and for $M > M^* \approx 2^{11.5}$ learning curves are similar to that of a model trained with infinite task diversity. **Right:** For $M \leq 2^{10}$, $t^*$ (the training step at which $\Delta_{\text{PT,Ridge}}^{\mathcal{T}_{\text{Pretrain}}}$ is minimized) is well modeled by a scaling law $t^* \propto M^\alpha$. A linear fit of $\log t^*$ vs $\log M$ (dashed red line) gives $\alpha \approx 0.47$. But for $M > 2^{10}$, $\Delta_{\text{PT,Ridge}}^{\mathcal{T}_{\text{Pretrain}}}$ decreases through training; $t^* = 2M$, is larger than predicted by the scaling law. This sudden break in the scaling law suggests a fundamental difference in the learning dynamics of models on either side of the threshold.

**Finite size scaling of training data suggests an algorithmic phase transition as task-diversity increases.** The experiments in Fig. 2 (left column) suggest that, when tested on both task distributions $\mathcal{T}_{\text{Pretrain}}$ and $\mathcal{T}_{\text{True}}$, the ICL algorithm implemented by a PT exhibits a smooth crossover in performance from dMMSE to Ridge. We next examine how this transition changes as we increase the number of sequences per task seen over pretraining, at fixed task diversity. One might reasonably expect that, if the transformer sees more sequences per latent task in $\mathcal{T}_{\text{Pretrain}}$, both its predictions and performance should become more similar to those of dMMSE, and less similar to those of Ridge, at all values of task diversity. Strikingly, this natural expectation is violated in a manner that facilitates ICL on $\mathcal{T}_{\text{True}}$.

At each number of tasks, we increase the number of sequences per task by increasing batch size from 256 to 512 to 1024, while leaving the number of training steps fixed at 500K. We observe that $\Delta_{\text{PT, dMMSE}}^{\mathcal{T}_{\text{Pretrain}}}$, which quantifies how different the PT and dMMSE estimator's predictions are when testing on tasks drawn from $\mathcal{T}_{\text{Pretrain}}$, does in fact decrease for $M \leq 2^{10}$ (Fig. 2 top center) as we train on more sequences per task. Moreover, for each $M \in \{2^{10}, ..., 2^{14}\}$ the PT's predictions also become less similar to those of Ridge, both on tasks from $\mathcal{T}_{\text{Pretrain}}$ (Fig. 2, top right) and $\mathcal{T}_{\text{True}}$ (Fig. 2, bottom right). **Crucially**, this movement in behavior of the PT towards dMMSE and away from Ridge, at least on tasks drawn from $\mathcal{T}_{\text{Pretrain}}$, holds *only* up to a threshold number of tasks between $2^{14}$ and $2^{15}$. Beyond this threshold, pretraining on more sequences per task at a fixed task diversity actually makes the PT *more* like Ridge, in that both $\Delta_{\text{PT,Ridge}}^{\mathcal{T}_{\text{Pretrain}}}$ and $\Delta_{\text{PT,Ridge}}^{\mathcal{T}_{\text{True}}}$ decrease (Fig. 2, right top and right bottom respectively). This means that, beyond a task diversity threshold, the PT can not only optimally solve new tasks from $\mathcal{T}_{\text{True}}$ by matching Ridge performance, but also the PT *gets better* at doing so if trained on more sequences per task, despite the limited set of tasks experienced in pretraining. Thus, in contrast to the natural expectation stated above, more sequences per task does not promote overspecialization of the PT to the $\mathcal{T}_{\text{Pretrain}}$ at task diversities beyond the threshold.

Finally, the motion of the ICL algorithm implemented by PT towards (away) from Ridge above (below) a task diversity threshold (Fig. 2, right top and bottom) indicates that as one increases the number of sequences per task at fixed task diversity, the smooth cross over in performance of the PT between dMMSE and Ridge, shown in Fig. 2, left top and bottom, will become sharper and sharper in task diversity, ultimately exhibiting a sharp phase transition in the limit of infinite number of sequences per task. Remarkably, this phase transition in the ICL algorithm implemented by the PT appears at a moderate task diversity threshold below $2^{15}$ pretraining tasks; even though dMMSE significantly underperforms relative to Ridge on $\mathcal{T}_{\text{True}}$ at this task diversity, the PT nevertheless remains unimpaired by this limited task diversity and can optimally solve new tasks.

**Increased training time at fixed batch size further supports an algorithmic phase transition.** To confirm the above results, we also increase the number of sequences per task, at each task diversity, by increasing the number of training steps $N$ from 500K to 1M while keeping batch size fixed at 256. We observe that doubling $N$ (change from pale blue to red in Fig. 3) and doubling $B$ (change from pale

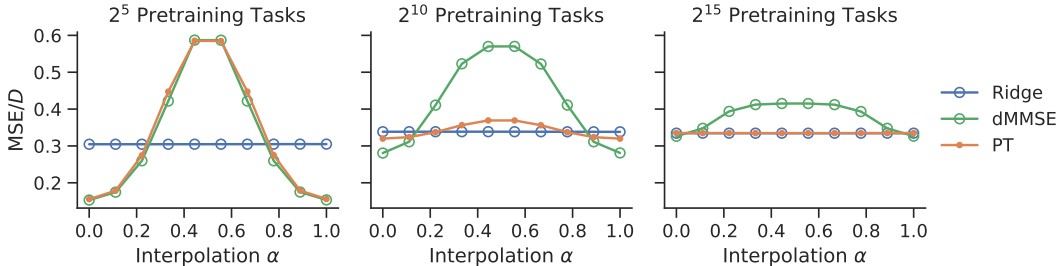

Figure 5: **Transformers pretrained with high, but not low, task diversity can learn *new* tasks in-context.** We compare the normalized loss of the PT to that of dMMSE and Ridge as we interpolate between tasks in the pretraining dataset. *Left*: At $2^5$ tasks, well below the task diversity threshold, the PT performance matches that of the dMMSE estimator along interpolating paths, but under-performs Ridge on *new* tasks at the center. *Middle*: At $2^{10}$ tasks, the PT outperforms dMMSE on *new* tasks at the center of the interpolation path, but is not yet as good as Ridge on new tasks. *Right*: At $M = 2^{15}$ tasks, just above the task diversity threshold, the PT performs as well as Ridge even on new tasks at the center. This demonstrates that, when pretrained on data with a finite but large number of unique tasks, the PT, unlike the Bayes optimal estimator for $\mathcal{T}_{\text{Pretrain}}$, can learn new tasks in-context.

blue to red in Fig. 2) have very similar effects on $\Delta_{\text{PT,dMMSE}}^{\mathcal{T}}$ and $\Delta_{\text{PT,Ridge}}^{\mathcal{T}}$, for both $\mathcal{T} = \mathcal{T}_{\text{True}}$ and $\mathcal{T} = \mathcal{T}_{\text{Pretrain}}$. More importantly, the task diversity threshold, which we determined as the cross-over point in $\Delta_{\text{PT,Ridge}}^{\mathcal{T}_{\text{True}}}$ between batch sizes 256, 512, and 1024 at 500K training steps (Fig. 2 bottom right) happens at the same number of tasks as the crossover point between 500K and 1M steps at batch size 256 (Fig. 3, right). Given that our two approaches for training the baseline transformer on more data yield the same task diversity threshold, and that doubling batch size leads to significantly faster training times than doubling number of steps, from here onward we consider the task diversity threshold to be cross-over point in $\Delta_{\text{PT,Ridge}}^{\mathcal{T}_{\text{True}}}$ between batch sizes 256 and 512 when training for 500K steps. See Appendix D for more ablations of batch size and training steps that provide further evidence for how the *number of sequences* seen by the transformer is the key factor determining the similarity of its predictions to those of dMMSE and Ridge at each number of tasks.

**Learning dynamics and a break in the scaling of early stopping time further supports an algorithmic phase transition.** To probe if the observed transition is merely an effect of under-fitting, we study the learning dynamics of *small* PTs in the *very large* number of steps regime. First, in Appendix E, we verify that the *small* PT also demonstrates an algorithmic phase transition but at lower task diversity threshold between $2^{11}$ and $2^{12}$ pretraining tasks. In Fig. 4 left, we visualize the learning curves ($\Delta_{\text{PT,Ridge}}^{\mathcal{T}_{\text{True}}}$ vs training steps) of PTs trained for 500K steps at batch size 512 with pretraining task diversities, $M$, below and above the task diversity threshold, $M^*$. For $M < M^*$, $\Delta_{\text{PT,Ridge}}^{\mathcal{T}_{\text{True}}}$ decreases early in training until it reaches a minimum at time step $t^*$, and then increases as the PT approaches dMMSE. We define $t^*$ as the early stopping time for Ridge. For $M > M^*$, $\Delta_{\text{PT,Ridge}}^{\mathcal{T}_{\text{True}}}$ decreases throughout training. To evaluate if, in the latter case, models are undertrained and $t^*$ is larger than the total training time, we extend training to 2M steps at batch size 512 ($4\times$ the training time, see Appendix B). Fig. 4 center, shows these learning curves along with that of the model trained with infinite task diversity; even in this long training regime, the task diversity threshold does not change. For both short and long training durations, models trained with the same $M$ have similar qualitative behavior (whether distance to Ridge decreases then increases or monotonically decreases). Additionally, learning curves of the models with $M > M^*$ are very similar to the learning curves for models trained on infinite pretraining task diversities and they achieve similar final accuracy (dahed lines vs markers in Fig. 10), suggesting that these models are approaching the Ridge solution.

In Fig. 4 right, we study how $t^*$, scales with $M$. For most $M < M^*$, $t^*$ obeys a simple scaling behavior $t^* \propto M^\alpha$, $\alpha \approx 0.47$. However, for $M > 2^{10}$, the distance to Ridge decreases monotonically through training and $t^* = 2M$ steps. Despite the caveat that our experiments are necessarily in the large but *finite* training step regime with a decayed learning rate schedule, this stark break in the scaling behavior of $t^*$ near the task diversity threshold suggests that the observed transition is not just caused by under-fitting but an underlying difference in the learning dynamics.

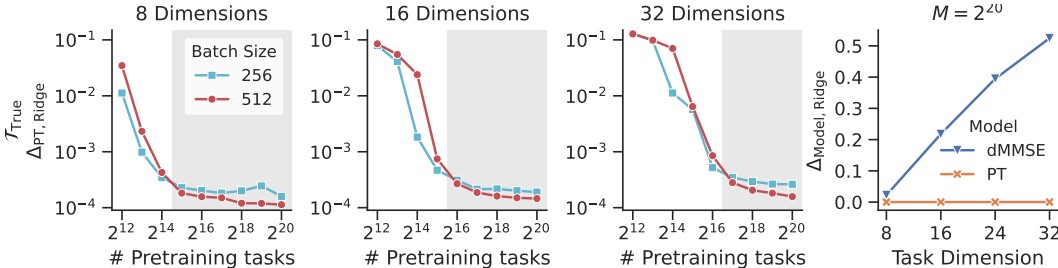

Figure 6: **The task diversity threshold increases with task dimension, and the PT's ability to solve new tasks scales significantly better than dMMSE's.** We vary the dimension of the regression problem $D$ (*first three panels*) while leaving the signal-to-noise ratio fixed. The task diversity threshold consistently increases with task dimension (gray shading denotes post threshold). At $2^{20}$ tasks (*right*), the PT's predictions are similar to those of Ridge at all $D$ (orange $\Delta_{\text{PT,Ridge}}^{\mathcal{T}_{\text{True}}}$), whereas dMMSE grows progressively less similar to Ridge (blue $\Delta_{\text{Ridge,dMMSE}}^{\mathcal{T}_{\text{True}}}$).

**The transition along interpolating paths.** To obtain an additional description of the algorithmic transition in the PT from dMMSE to Ridge, we compute the ICL performance of the PT, and compare it to both dMMSE and Ridge, on a one parameter family of new tasks $\mathbf{w}_\alpha$ that interpolate between pairs of seen tasks $\mathbf{w}_i$ and $\mathbf{w}_j$ in the support of $\mathcal{T}_{\text{Pretrain}}$. The interpolation path is given by

$$\mathbf{w}_\alpha = \frac{1}{2}(\|\mathbf{w}_i\|_2 + \|\mathbf{w}_j\|_2)\frac{\alpha\mathbf{w}_i + (1-\alpha)\mathbf{w}_j}{\|\alpha\mathbf{w}_i + (1-\alpha)\mathbf{w}_j\|_2} \qquad \text{for } \alpha \in [0,1]. \qquad (5)$$

Here we fix the norm of the interpolated vector $\mathbf{w}_\alpha$ to the average of the two endpoint norms to avoid $\|\mathbf{w}_\alpha\|$ taking on very small values for $\alpha \sim \frac{1}{2}$. Fig. 5 shows the results of this analysis for $2^5$ (left, low task diversity regime), $2^{10}$ (center, below task diversity threshold), and $2^{15}$ (right, just above the task diversity threshold) tasks. At each value of $\alpha$, MSE is averaged over a large number of task pairs $(\mathbf{w}_i, \mathbf{w}_j)$. Examination of the average performance at the center of many interpolation paths, corresponding to fundamentally *new* tasks far from tasks seen during pretraining, clearly reveals a transition in PT performance from dMMSE to Ridge, where new tasks can only be optimally learned above, but not below, the task diversity threshold. In contrast, unlike the PT, dMMSE cannot solve new tasks at any task diversity in the range considered.

**The PT outperforms a smoothed dMMSE model.** We have seen that at an intermediate task diversity the PT significantly outperforms dMMSE on new tasks in $\mathcal{T}_{\text{True}}$. It is clear why dMMSE performs poorly on new tasks in $\mathcal{T}_{\text{True}}$ at low task diversity: its prior over tasks concentrates on $M$ unique tasks in $\mathcal{T}_{\text{Pretrain}}$, while the prior over tasks in $\mathcal{T}_{\text{True}}$ is Gaussian. A natural conjecture is that the PT cannot memorize all $M$ tasks in $\mathcal{T}_{\text{Pretrain}}$ for large enough $M$. Therefore we also compare PT performance to a smoothed dMMSE estimator in which the discrete point prior over $M$ tasks seen in pretraining is replaced with a mixture of $M$ isotropic Gaussians with the same centers but with variance chosen to optimize performance on $\mathcal{T}_{\text{True}}$ (see Appendix G for details). This smoothed dMMSE outperforms dMMSE as it has a prior over tasks closer to the Gaussian $\mathcal{T}_{\text{True}}$. But remarkably, the PT still outperforms the smoothed dMMSE even with optimal smoothing (Fig. 12). This indicates the PT, even at moderate task diversity, implements a more sophisticated algorithm than a simple smoothed dMMSE arising from the PT's inability to resolve the $M$ pretraining tasks to high precision.

### 3.2 The PT exhibits superior scaling of task diversity threshold with dimension than dMMSE.

We next explore the dependence of the task diversity threshold on the regression problem dimension $D$. We vary $D \in \{8, 16, 32\}$ while simultaneously scaling up maximal context length as $K = 2D$, and increasing observation noise $\sigma^2$ to match the SNR to that of $D = 8$ and $\sigma^2 = 0.25$. We also train a larger model with 12 layers, 256-dimensional embeddings, and 4 attention heads that is sufficiently expressive to match Ridge performance at $D = 32$. Fig. 6, first 3 panels reveal that the task diversity threshold of the PT increases moderately (approximately linearly) with task dimension (i.e. roughly $2^{14}$, $2^{15}$, and $2^{16}$ at $D = 8, 16$, and $32$ respectively). This linear scaling is remarkable considering the volume of all possible tasks scales *exponentially* with dimension due to the concentration of the Gaussian $\mathcal{T}_{\text{True}}$ to a sphere for large $D$. Thus we expect dMMSE performance to scale much more poorly with dimension $D$ since the finite number of tasks in $\mathcal{T}_{\text{Pretrain}}$ would need to cover a substantial portion of the sphere for dMMSE to approach Ridge. To test this hypothesis, for $M = 2^{20}$, which is

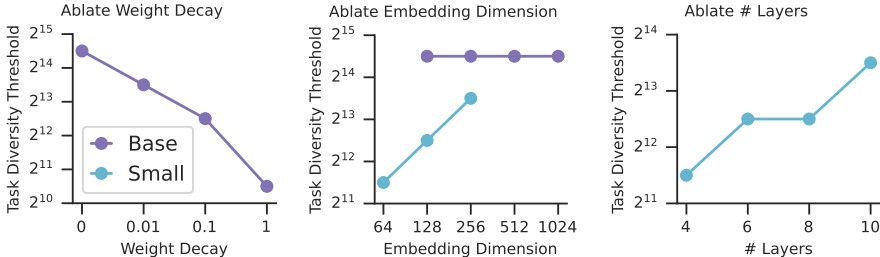

Figure 7: **Explicit regularization and model capacity affect the task diversity threshold.** Increasing explicit regularization, in the form of weight decay, lowers the task diversity threshold in *base* PTs (*left*). Increasing embedding dimension, has no effect on the threshold for *base* PTs, but does increase the threshold for a *small* PT (*middle*). Increasing depth, while holding other hyperparameters fixed, increases the threshold for a *small* PT (*right*). See Figure 13 for plots of $\Delta_{\text{dMMSE,Ridge}}^{\mathcal{T}_{\text{True}}}$ vs $M$.

the largest task diversity we consider, we explore how the similarity of PT and dMMSE predictions to Ridge on new tasks scales with $D$ (Fig. 6, right panel). We see that $\Delta_{\text{dMMSE,Ridge}}^{\mathcal{T}_{\text{True}}}$ grows significantly as we increase $D$, while remarkably $\Delta_{\text{PT,Ridge}}^{\mathcal{T}_{\text{True}}}$ is largely dimension independent. Overall this indicates that the scaling of PT error with dimension is vastly superior than that of dMMSE; PT remains near optimal and close to Ridge at all $D$ for $M = 2^{20}$, while dMMSE departs from Ridge as $D$ increases.

### 3.3 Effect of Regularization and model capacity on the task diversity threshold.

We study the dependence of the task diversity threshold on various hyperparameters. First, adding explicit regularization in the form of weight decay (see Appendix B for details), and increasing its value over three orders of magnitude, consistently lowers the threshold task diversity (Fig. 7, left). Note however, the lower task diversity threshold also comes with worse performance (Figure 13, top). This suggests that various forms of *implicit* regularization could help drive the algorithmic transition in the PT without weight decay. We also explore the effect of model capacity on the task diversity threshold by either increasing the embedding dimension of both *small* and *base* PTs or increasing the depth of *small* PTs. Fig. 7 center shows that increasing embedding dimension over a reasonable range does not affect the task diversity threshold of *base* PT. However, for *small* PT, increasing either the embedding dimension (Fig. 7 center) or depth (Fig. 7 right) increases the task diversity threshold. *Base* PT has a much larger capacity then *small* PT and also has a larger threshold; we hypothesize that *small* PT is still in a regime where the threshold is sensitive to capacity while *base* PT is not. Together, these results suggest that model capacity plays an important role in the emergence of in-context learning: increasing capacity (up to a point) leads to an increase in the task-diversity threshold.

## 4 Related work

The Bayesian framework for ICL introduced by Xie et al. [3], which motivates our work, hypothesizes that PTs "locate" concepts learned during pretraining to solve ICL tasks. A series of empirical work in language models [18–20] use this framework to select better in-context examples while Min et al. [21] use it to study the robustness of latent task inference. Our work builds on this framework in the linear regression setting and validates it at low task diversities. However, we find a regime—large but finite number of pretraining tasks—in which the ability to learn new tasks in-context is an emergent phenomenon that cannot be fully explained by Bayesian inference.

Prior work [12–14] has also shown that transformers can do linear regression in-context. However, they pretrain with unlimited task diversity, sampling a completely new regression vector for each sequence. In contrast, our work considers pretraining datasets with limited task diversity where ICL on new tasks emerges even though the pretraining loss does not explicitly encode it. Another line of work hypothesizes that ICL performs gradient descent in the activations of the forward pass, providing explicit constructions for the weights of the PT to implement this for linear regression [13, 14] or exploring this hypothesis in language models [22]. However more experiments are required to test the hypothesis that *trained* transformers actually match proposed constructions. Instead of studying the explicit mechanism by which in-context learning is implemented, our work focuses on the impact of the pretraining task diversity. Similar questions pertaining to the role of task diversification have been explored in the meta-learning literature [23–25].

Kirsch et al. [26] also show the emergence of in-context learning with pretraining task diversity on a toy classification task. By studying this question in the controlled setting of linear regression, we can compare to the optimal estimators on $\mathcal{T}_{\text{Pretrain}}$ and $\mathcal{T}_{\text{True}}$. This allows us to establish that ICL at finite task diversity emerges because the PT *departs* from the optimal estimator on the pretraining task distribution, and is not just a consequence of the pretraining task distribution becoming similar to the ideal task distribution. Among other important perspectives on ICL, Chan et al. [27] identify, in a toy setting, several properties of the training distribution—burstiness and occurrence of rare classes—that are necessary for the emergence of ICL. Wei et al. [28] study how ICL in large language models is affected by semantic priors and input-label mappings, focusing on differences across model scale. Olsson et al. [29] study inductions heads—circuits responsible for completing patterns by copying tokens—as a mechanism for implementing ICL.

## 5 Discussion

Overall, we have extensively explored the impact of pretraining task diversity on the emergence of in-context learning of fundamentally *new* tasks not seen during pretraining. We found several surprises by working in the controlled setting of linear regression, where we could compare the performance of the PT to Bayesian estimators that are optimal, either for the limited diversity pretraining task distribution $\mathcal{T}_{\text{Pretrain}}$ (i.e. dMMSE), or for the diverse ideal task distribution $\mathcal{T}_{\text{True}}$ (i.e. Ridge). These comparisons reveal an algorithmic phase transition in the PT from the former to the latter at an intermediate task diversity threshold; beyond this threshold, the PT solves fundamentally new tasks not seen during pretraining. Strikingly, this task diversity threshold scales moderately with task dimension, over the range of dimensions considered, despite the exponential growth in the volume of all possible tasks with dimension. Indeed this PT scaling vastly outperforms that of dMMSE. Overall, these results indicate that ICL of new tasks by PTs is an emergent phenomenon that cannot be explained by Bayesian inference on limited diversity pretraining task distributions. Moreover, our experiments suggest some form of implicit regularization in PTs allows them to break free of the pretraining task distribution to solve new tasks, given a moderate pretraining task diversity.

Remarkably, beyond the task diversity threshold, PTs learn the *optimal* estimator for the underlying generative model for pretraining tasks; this is the case for both Gaussian and Laplace priors over tasks (see Figure 14 for experiments with Laplace prior). This is true even though solutions with lower training loss exist; indeed when trained on more data at fixed diversity, PTs behave more like Ridge at the expense of higher training loss. Our experiments in Fig. 4 suggest that this algorithmic transition is due an underlying change in learning dynamics. We explore this hypothesis by probing the *linear mode connectivity* of the loss landscape [30, 31]. In Fig. 11 we find that PTs trained with large $M$ inhabit the same loss basin as PTs trained with $M = \infty$: the training loss barrier between PTs trained with $M \gtrsim 2^{13}$ and PTs with $M = \infty$ is similar to two PTs trained with $M = \infty$. In contrast, there are large loss barriers between PTs trained with $M < 2^{13}$ and $M = \infty$. Additionally, PTs trained with $M = \infty$ are closer in weight space to PTs trained with large $M$ than those trained with small $M$ (see Appendix F). Overall, these experiments provide further evidence that PTs trained with task diversities beyond the threshold find solutions similar to the optimal model for $\mathcal{T}_{\text{True}}$; we leave further exploration of these loss landscapes to future work.

An intriguing question is how these observations carry over to language. A key mystery about the efficacy of ICL in language tasks lies in how different the tasks learned in-context are from the pretraining distribution of large language corpora. It is also less clear how to categorize the contents of such corpora according to tasks and measure their resulting task diversity. Regardless, our observation in linear regression that a moderate threshold in pretraining task diversity can enable PTs to solve new tasks may imply that many language tasks that are quite different from the statistics of large language corpora can still nevertheless be solved in-context.

Our results also suggest that the scale of data alone does not lead to good ICL performance. In fact, below the task diversity threshold, increasing the size of the pretraining dataset without increasing task diversity hurts ICL performance. It is necessary to increase both the diversity and size of the dataset for ICL to emerge. Thus to improve ICL in language settings, our work motivates future studies into uncovering the relevant notion of tasks in language modeling and approaches to increase task diversity in language corpora. More generally, our empirical analysis of the impact of pretraining task diversity on ICL motivates further theoretical studies. Such studies will be key to understanding the mystery of why simple next token prediction during pretraining can lead to in-context learning of so many apparently different tasks.

## Acknowledgements

The authors would like to thank the Google TPU Research Cloud (TRC) whose generous support made this project possible. SG thanks an NSF CAREER award and NTT Research for support.

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

# A  Bayesian estimators

## A.1  Bayesian MMSE estimator

The optimal estimator for the $k$th prediction, $\hat{y}_k^{\mathcal{T}}$, that minimizes the $k$th term in the sum in $L^{\mathcal{T}}$ (in Eq. (1)) is the Bayesian estimator with $\mathcal{T}$ as the prior. This is given by the posterior mean of $y_k$ conditioned on the context: $\hat{y}_k^{\mathcal{T}} = \mathbb{E}_{\mathcal{T},\varepsilon_k}[y_k \mid S_k]$, where the expectation is over the task distribution, $\mathcal{T}$, and the noise, $\varepsilon_k$. To derive this, notice that, using the law of total expectation, the $k$th term in the loss $L^{\mathcal{T}}$ is:

$$\mathop{\mathbb{E}}_{\substack{\mathbf{w}\sim\mathcal{T} \\ \mathbf{x}_1,...,\mathbf{x}_k\sim\mathcal{N}(\mathbf{0},\mathbf{I}_D) \\ \varepsilon_1,...,\varepsilon_k\sim\mathcal{N}(0,\sigma^2)}} (f_\theta(S_k) - y_k)^2 = \mathop{\mathbb{E}}_{S_k} \mathop{\mathbb{E}}_{\substack{\mathbf{w}\sim\mathcal{T} \\ \varepsilon_k\sim\mathcal{N}(0,\sigma^2)}} \left[(f_\theta(S_k) - y_k)^2 \Big| S_k\right]$$

which implies that the optimal estimator is $\hat{y}_k^{\mathcal{T}} = \mathbb{E}_{\mathcal{T},\varepsilon_k}[y_k \mid S_k]$. Adopting the notation $\mathbf{X} = (\mathbf{x}_1^\mathsf{T}, ..., \mathbf{x}_{k-1}^\mathsf{T}) \in \mathbb{R}^{(k-1)\times D}$ and $\mathbf{y} = (y_1, ..., y_{k-1})$, we can express the expectation explicitly:

$$\mathbb{E}[y_k \mid S_k] = \int d\mathbf{w}dy_k\, y_k p(\mathbf{w}, y_k \mid \mathbf{X}, \mathbf{y}, \mathbf{x}_k)$$

$$= \int d\mathbf{w}dy_k\, y_k p(y_k \mid \mathbf{x}_k, \mathbf{w})p(\mathbf{w} \mid \mathbf{X}, \mathbf{y})$$

$$= \int d\mathbf{w}\, \mathbf{w}^\mathsf{T}\mathbf{x}_k\, p(\mathbf{w} \mid \mathbf{X}, \mathbf{y})$$

$$\equiv \hat{\mathbf{w}}_k^\mathsf{T}\mathbf{x}_k$$

where

$$\hat{\mathbf{w}}_k = \mathbb{E}[\mathbf{w} \mid S_k] = \frac{\int d\mathbf{w}p(\mathbf{w})\mathbf{w}\prod_{i=1}^{k-1}p(y_i|\mathbf{x}_i,\mathbf{w})}{\int d\mathbf{w}p(\mathbf{w})\prod_{i=1}^{k-1}p(y_i|\mathbf{x}_i,\mathbf{w})} \tag{6}$$

## A.2  dMMSE estimator

For task distribution $\mathcal{T}_{\text{Pretrain}} = \mathcal{U}\{\mathbf{w}^{(1)}, ..., \mathbf{w}^{(M)}\}$, we can directly get the discrete minimum mean squared error **(dMMSE)** estimator by plugging the uniform discrete distribution into Eq. 6.

$$\hat{\mathbf{w}}_k^{\text{dMMSE}} = \sum_{i=1}^M \frac{\exp\left(-\frac{1}{2\sigma^2}\sum_{j=1}^{k-1}(y_j - \mathbf{w}^{(i)\mathsf{T}}\mathbf{x}_j)^2\right)}{\sum_{l=1}^M \exp\left(-\frac{1}{2\sigma^2}\sum_{j=1}^{k-1}(y_j - \mathbf{w}^{(l)\mathsf{T}}\mathbf{x}_j)^2\right)}\mathbf{w}^{(i)}. \tag{7}$$

## A.3  Ridge estimator

For task distribution $\mathcal{T}_{\text{True}} = \mathcal{N}(\mathbf{0}, \mathbf{I}_D)$, we can directly get the Ridge regression estimator from Eq. 6:

$$\hat{\mathbf{w}}_k^{\text{Ridge}} = \frac{\int \mathbf{w}\exp\left[-\frac{1}{\sigma^2}(\mathbf{X}\mathbf{w} - \mathbf{y})^T(\mathbf{X}\mathbf{w} - \mathbf{y}) - \mathbf{w}^T\mathbf{w}\right]d\mathbf{w}}{\int \exp\left[-\frac{1}{\sigma^2}(\mathbf{X}\mathbf{w} - \mathbf{y})^T(\mathbf{X}\mathbf{w} - \mathbf{y}) - \mathbf{w}^T\mathbf{w}\right]d\mathbf{w}}$$

$$= \frac{\int \mathbf{w}\exp\left[-\frac{1}{\sigma^2}\left(\mathbf{w} - (\mathbf{X}^T\mathbf{X} + \sigma^2\mathbf{I}_D)^{-1}\mathbf{X}\mathbf{y}\right)^T(\mathbf{X}^T\mathbf{X} + \sigma^2\mathbf{I}_D)\left(\mathbf{w} - (\mathbf{X}^T\mathbf{X} + \sigma^2\mathbf{I}_D)^{-1}\mathbf{X}\mathbf{y}\right)\right]d\mathbf{w}}{\int \exp\left[-\frac{1}{\sigma^2}\left(\mathbf{w} - (\mathbf{X}^T\mathbf{X} + \sigma^2\mathbf{I}_D)^{-1}\mathbf{X}\mathbf{y}\right)^T(\mathbf{X}^T\mathbf{X} + \sigma^2\mathbf{I}_D)\left(\mathbf{w} - (\mathbf{X}^T\mathbf{X} + \sigma^2\mathbf{I}_D)^{-1}\mathbf{X}\mathbf{y}\right)\right]d\mathbf{w}}$$

$$= \left(\mathbf{X}^\mathsf{T}\mathbf{X} + \sigma^2\mathbf{I}_D\right)^{-1}\mathbf{X}^\mathsf{T}\mathbf{y} \tag{8}$$

where $\mathbf{X} = (\mathbf{x}_1^\mathsf{T}, ..., \mathbf{x}_{k-1}^\mathsf{T}) \in \mathbb{R}^{(k-1)\times D}$ and $\mathbf{y} = (y_1, ..., y_{k-1})$.

# B  Experimental details

For most experiments, we study linear regression in $D = 8$ dimensions with up to $K = 16$ in-context examples and observation noise variance $\sigma^2 = 0.25$. Our base model is a transformer with the GPT2 architecture [15] with 8 layers, 128-dimensional embeddings, and 2 attention heads. We train with

the Adam optimizer [16] and a one-cycle triangle learning rate schedule [17] with 50% warmup. The base model is trained with batch size 256 for 500K training steps, though these hyperparameters are varied in our experiments. Specifically, for the experiments in Fig. 3 we sweep over 500K and 1M training steps to find the task diversity threshold as a function of training steps. For all other experiments, we sweep batch size 256 and 512 and for the experiments in Fig. 2, we do an additional batch size of 1024.

For the experiments in Fig. 6, we use a larger model with 12 layers, 256-dimensional embeddings, and 4 attention heads. This ensures that the model can learn to perform ridge regression in the $D = 32$ setting. We also train the models on $D = 8, 16, 24, 32$ with up to $K = 2D$ in-context examples in each case. The observation noise variance is scaled at each dimension to keep the signal to noise ratio constant. So $\sigma^2 = 0.5, 0.707, 0.866, 1$ for $D = 8, 16, 24, 32$ respectively.

We always sweep over learning rates in $\{0.0001, 0.0003, 0.001\}$ and choose the largest learning rate at which training is stable. For almost all experiments, the learning rate is 0.001, the main exception being the experiments in Fig. 6 where the learning rate is 0.0001.

For the 2M training step experiments in Fig. 4 and Fig. 10, the learning rate increases linearly up to 0.001 at 250K steps (i.e. using the same warmup as in the 500K step experiments) and then decreases linearly to 0 at 2M steps, consistent with our choice of decayed learning rate schedules. The experiments in Appendix F are performed at batch size 512 and following the 500K step one-cycle triangle learning rate schedule described above.

The implementation for this paper was done in JAX and all experiments were run on TPU v2-8 and v3-8 provided as part of the Google TPU Research Cloud program. Each training run takes approximately 4 v2-8 TPU hours.

## C   Support Figure 2

Fig. 8 provides an additional visualization for Fig. 2 upper middle but on a linear scale on the y-axis. In this figure it is clear that for all task diversities below the threshold—less that $2^{14}$—increasing the batch size aligns the PT with dMMSE, the optimal estimator for $\mathcal{T}_{\text{Pretrain}}$. Thus below the task diversity threshold, the optimal estimator does indeed approach the Bayesian estimator with prior $\mathcal{T}_{\text{Pretrain}}$ when trained on more data and is thus unable to learn new tasks as discussed in Section 3.1.

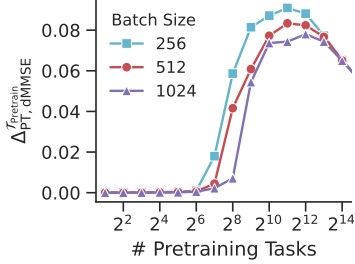

Figure 8: Another visualization for Fig. 2 upper middle but with the x-axis extending all the way to the task diversity threshold and a linear scale on the y-axis to make it easy to differentiate the different batch sizes at large number of pretraining tasks. See Appendix C for details.

## D   Dependence on number of sequences per task

In Section 3.1 we explore how the crossover of the PT from behaving like dMMSE to behaving like Ridge as we increase pretraining task diversity depends on the number of sequences per task seen during pretraining. In Fig. 2 middle and right column, we probe this by increasing the batch size to increase the number of sequences per task seen during pretraining at each level of task diversity. To ensure that that the phase transition and task diversity threshold we find is indeed a consequence of increasing the number of sequences per task and not merely just an effect of increasing batch size, we keep the batch size constant and increase the number of sequences per task in Fig. 3.

In Fig. 9 we provide additional evidence that the number of sequences per task and not batch size nor number of training steps is the primary factor that drives the behavior of the final model. Specifically, we show that the final behavior of the model is invariant to batch size or number of training steps if the number of training sequences per task is kept constant, atleast within a reasonable range of batch size and number of training steps. To do so, at each level of pretraining task diversity, we compare our base model (batch size = 256, 500K training steps, red cicles) to a model trained with batch size 512 (light blue squares) and batch size 128 (purple triangles). For either variant of the batch size, we adjust the number of training steps to keep the total number of pretraining sequences per task constant: 250K training steps for batch size 512 and 1M training steps for batch size 128. In all three cases, the behavior of the PT is identical as measured by $\Delta^{\mathcal{T}}_{\text{PT,dMMSE}}$ and $\Delta^{\mathcal{T}}_{\text{PT,Ridge}}$ for both $\mathcal{T}_{\text{Pretrain}}$ and $\mathcal{T}_{\text{True}}$ across all numbers of pretraining tasks.

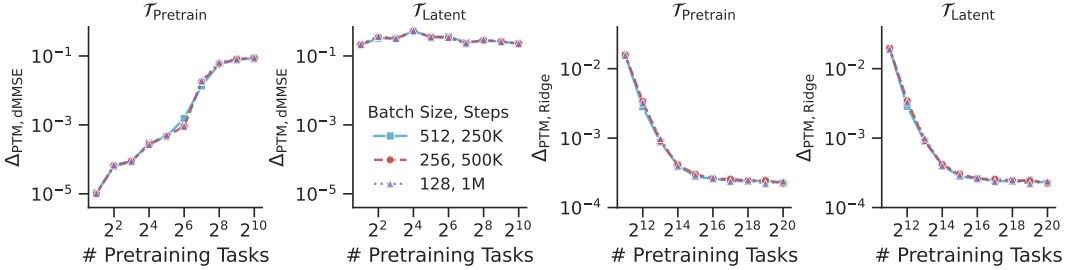

Figure 9: **Number of sequences per task and not batch size or number of training steps affects final model behavior**: This figure provides additional experiments to support the findings in Section 3.1 and Figs. 2 and 3. Each panel corresponds to the same panel in Fig. 3. See Appendix D for details.

# E   Small PT

We run our task diversity threshold finding experiment in a *small* PT with 64 dimensional embeddings, 4 layers, and 2 attention heads. The existence of a task diversity threshold reproduces in a Small PT: the PTs transition from becoming less like Ridge to becoming more like Ridge when trained on more data (see Fig. 10). We also note that the Small PT with a lower capacity has a lower threshold ($\sim 2^{11.5}$ compared to $\sim 2^{14.5}$ for the Base PT in Fig. 2) but also has lower overall performance (compare y-axis range to Fig. 2 lower right).

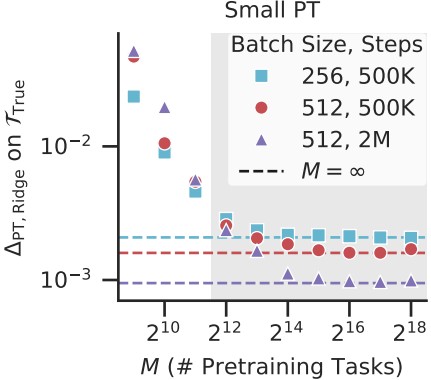

Figure 10: **Existence of a task diversity threshold reproduces in *small* PT.** Starting from batch size 256 and 500K training steps, we train on 2x data (i.e. examples per task) by doubling the batch size, and on 8x data by doubling batch size and quadrupling the number of training steps. We show distance to Ridge on new tasks (as in Fig. 2 lower right), and also plot distance to Ridge for models trained with infinite task diversity, where each training example samples a new task from the Gaussian distribution, as dashed lines (colors indicating batch size and training time). Points after the threshold ($M \geq 2^{12}$) are shaded gray. Importantly, the threshold value doesn't change even if we go from 2x to 8x data suggesting that it isn't an artifact of underfitting.

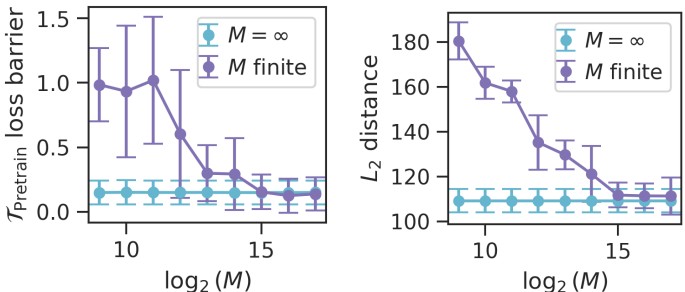

Figure 11: **The training loss barrier between a PT trained with** $M \gtrsim 2^{13}$ **and a PT trained with** $M = \infty$ **is small.** At each $M$, including $M = \infty$, we train four *small* PTs, each with a different random seed for tasks. Each purple dot is an average over the 16 finite-infinite task model pairs at the given value of $M$, and each light blue dot is an average over the 6 infinite-infinite task model pairs. Error bars are standard deviations. See Appendix F for details on how loss barriers and distances between models are computed.

## F    Examining the loss landscape along paths interpolating between finite and infinite task models

For each finite number of tasks, $M$, as well as $M = \infty$ (where each training example samples a new task from the Gaussian distribution), we train four small PTs, each with a different random seed *for tasks*. This means that all randomness (including model initialization, data, and observation noise) is shared across the four experiments, *except* for that governing the sampling of $\mathbf{w}$'s used for training. We then compare the finite $M$ models to the $M = \infty$ models, both in terms of the size of the training loss barrier along a path interpolating between the models, as well as the distance in weight space between the models.

We compute the training loss barrier between a finite $M$ model and an infinite task model under the objective $L_{\text{Pretrain}}^{\mathcal{T}}$ defined by the **specific set of** $M$ **tasks** the finite task model was trained on. We follow the approach outlined in [31] where the barrier between two models is the loss at a model whose weights are the average of the two models, minus the average of the losses of the two models. Specifically, given two weight configurations $w_1$ and $w_2$ and a loss function $L$, the loss barrier is calculated as

$$\text{loss barrier} = L\left(\frac{w_1 + w_2}{2}\right) - \frac{L(w_1) + L(w_2)}{2} \tag{9}$$

The distance in weight space, in turn, is simply the $L_2$ norm of the difference in model weights. For each $M$ we average barriers and distances across all finite-infinite task pairs of models. We treat the analogous quantities computed between pairs of infinite task models as a baseline. See Fig. 11 and the Discussion section in the main text.

## G    Smoothed dMMSE

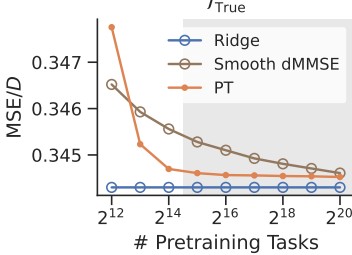

Figure 12: **The PT outperforms the Smoothed dMMSE estimator with optimal smoothing.** See Appendix G.

Here we consider the setting in which the discrete point prior over $M$ tasks seen in pretraining is replaced with a mixture of $M$ isotropic Gaussians with variance $\epsilon^2$. We call the estimator that does posterior predictive inference under this prior the Smoothed dMMSE estimator (sMMSE). In the limit of $\epsilon \to 0$, the sMMSE estimator becomes the dMMSE estimator. On the other hand, if $\epsilon \to \infty$, the prior will look like a Gaussian distribution but will have a variance much larger than the variance of $\mathcal{T}_{\text{True}}$. In both edge cases, it won't perform well on $\mathcal{T}_{\text{True}}$, and so there is an optimal $\epsilon$ that achieves the best performance on $\mathcal{T}_{\text{True}}$. In practice, we perform a search by simulation to determine the optimal $\epsilon$. We observe in Fig. 12 that for $2^{13}$ tasks onward, the PT outperforms the optimally smoothed sMMSE on $\mathcal{T}_{\text{True}}$.

To derive the sMMSE estimator, we want to calculate $\hat{\mathbf{w}}_k = \mathbb{E}[\mathbf{w} \mid S_k]$ and apply Eq. (6).

$$
\begin{aligned}
p(\mathbf{w}|S_k) &= \frac{p(S_k|\mathbf{w})p(\mathbf{w})}{p(S_k)} \\
&\propto \prod_{j=1}^{k-1} p(y_j|\mathbf{x}_j, \mathbf{w}) \sum_{i=1}^{M} \exp\left(-\frac{1}{2\epsilon^2}\left(\mathbf{w} - \mathbf{w}^{(i)}\right)^2\right) \\
&\propto \sum_{i=1}^{M} \exp\left(-\frac{1}{2\sigma^2} \sum_{j=1}^{k-1} (\mathbf{w}^\mathsf{T} \mathbf{x}_j - y_j)^2 - \frac{1}{2\epsilon^2}(\mathbf{w} - \mathbf{w}^{(i)})^2\right) \\
&\propto \sum_{i=1}^{M} \exp\left(-\frac{1}{2}\mathbf{w}^\mathsf{T}\left(\frac{1}{\sigma^2}\mathbf{X}^\mathsf{T}\mathbf{X} + \frac{1}{\epsilon^2}\mathbf{I}\right)\mathbf{w} + \left(\frac{1}{\epsilon^2}\mathbf{w}^{(i)} + \frac{1}{\sigma^2}\mathbf{X}^\mathsf{T}\mathbf{y}\right)^\mathsf{T}\mathbf{w} - \frac{1}{2\epsilon^2}\left\|\mathbf{w}^{(i)}\right\|^2\right)
\end{aligned}
$$

We define,

$$
\begin{aligned}
\mathbf{A} &= \frac{1}{\sigma^2}\mathbf{X}^\mathsf{T}\mathbf{X} + \frac{1}{\epsilon^2}\mathbf{I} \\
\mathbf{J}_i &= \frac{1}{\epsilon^2}\mathbf{w}^{(i)} + \frac{1}{\sigma^2}\mathbf{X}^\mathsf{T}\mathbf{y} \\
C &= \sum_{i=1}^{M} \exp\left(-\frac{1}{2\epsilon^2}\|\mathbf{w}^{(i)}\|^2\right) \int d\mathbf{w}\, \exp\left(-\frac{1}{2}\mathbf{w}^\mathsf{T}\mathbf{A}\mathbf{w} + \mathbf{J}_i^\mathsf{T}\mathbf{w}\right) \\
&= \sqrt{\frac{(2\pi)^D}{|\mathbf{A}|}} \sum_{i=1}^{M} \mathbf{J}_i^\mathsf{T}\mathbf{A}^{-1}\mathbf{J}_i \exp\left(-\frac{1}{2\epsilon^2}\|\mathbf{w}^{(i)}\|^2\right)
\end{aligned}
$$

so that,

$$
p(\mathbf{w} \mid S_k) = \frac{1}{C} \sum_{i=1}^{M} \exp\left(-\frac{1}{2}\mathbf{w}^\mathsf{T}\mathbf{A}\mathbf{w} + \mathbf{J}_i^\mathsf{T}\mathbf{w} - \frac{1}{2\epsilon^2}\|\mathbf{w}^{(i)}\|^2\right)
$$

Plugging back to Eq. (6),

$$
\begin{aligned}
\hat{\mathbf{w}}_k^{\text{sMMSE}} &= \frac{1}{C} \sum_{i=1}^{M} \exp\left(-\frac{1}{2\epsilon^2}\|\mathbf{w}^{(i)}\|^2\right) \int d\mathbf{w}\, \mathbf{w} \exp\left(-\frac{1}{2}\mathbf{w}^\mathsf{T}\mathbf{A}\mathbf{w} + \mathbf{J}_i^\mathsf{T}\mathbf{w}\right) \\
&= \frac{1}{C} \sum_{i=1}^{M} \sqrt{\frac{(2\pi)^D}{|\mathbf{A}|}} \mathbf{A}^{-1}\mathbf{J}_i \exp\left(\frac{1}{2}\mathbf{J}_i^\mathsf{T}\mathbf{A}^{-1}\mathbf{J}_i - \frac{1}{2\epsilon^2}\|\mathbf{w}^{(i)}\|^2\right) \\
&\equiv \sum_{i=1}^{M} \beta_i \mathbf{A}^{-1}\mathbf{J}_i
\end{aligned}
$$

where $\beta = \text{softmax}(\tilde{\beta})$ and $\tilde{\beta}_i \equiv \frac{1}{2}\mathbf{J}_i^\mathsf{T}\mathbf{A}^{-1}\mathbf{J}_i - \frac{\|\mathbf{w}^{(i)}\|^2}{2\epsilon^2}$.

# H   Supporting figures

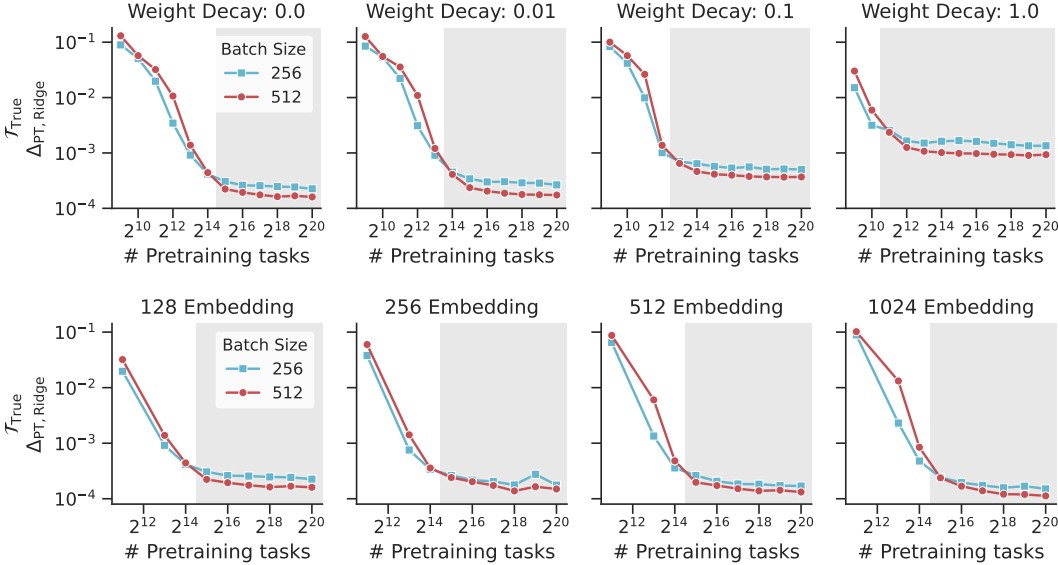

Figure 13: **Effect of weight decay and embedding dimension on the task diversity threshold of *base* PT.** Increasing explicit regularization, in the form of weight decay, consistently lowers the task diversity threshold, which is the crossover point between $\Delta_{\text{PT,Ridge}}^{\mathcal{T}_{\text{True}}}$ at the two batch sizes (*top*, gray shading denotes post threshold). However, note that more weight decay does make the PT's predictions *less* similar to those of Ridge, as quantified by $\Delta_{\text{PT,Ridge}}^{\mathcal{T}_{\text{True}}}$ at numbers of tasks beyond the task diversity threshold (i.e. the flat region of *both* blue and red curves progressively move up with increasing weight decay from left to right in the). Increasing embedding dimension, while scaling the number of heads by the same factor, has no discernible effect on the threshold (*bottom*) for a *base* PT.

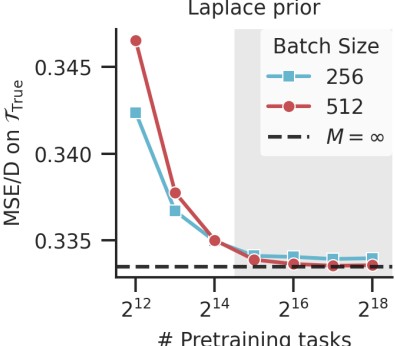

Figure 14: **Task diversity threshold for tasks sampled from a Laplace prior**. We run our task diversity threshold finding experiment in a setting where each element of each task regression vector is sampled from a Laplace prior instead of a Gaussian prior. We use the base model from our paper. For the baseline, we approximate the Bayes-optimal model on new tasks by training for 500K steps with batch size 512 on a pretraining distribution with infinite task diversity drawn from the Laplace prior (black dashed line). The y-axis is the dimension normalized mean squared error on new tasks drawn from the Laplace prior. In this setting, we see a task diversity threshold between $2^{14}$ and $2^{15}$ pretraining tasks; beyond the threshold (gray shading), training on more data leads to better performance on new, unseen tasks and ICL emerges. Thus, our findings for the task diversity threshold reproduce when tasks are drawn from different priors.

