# OpenReview forum: "Pretraining task diversity and the emergence of non-Bayesian in-context learning for regression"
_NeurIPS.cc/2023/Conference — NeurIPS 2023 poster_

### Official Review · Reviewer_CEjq · 2023-06-11

**Soundness:** 4 excellent
**Presentation:** 4 excellent
**Contribution:** 3 good
**Rating:** 8
**Confidence:** 4

**Summary:**

This paper empirically studies whether the ability of transformers to perform in-context learning can be explained by Bayesian inference in a setting wherein tasks are linear regression problems. The pre-training task distribution is the uniform distribution over M unique tasks (ground-truth regressors), and the population distribution of tasks is the standard Gaussian distribution over ground-truth regressors in $R^D$. The experiments shed light on a number of surprising phenomena, all centered around the observation that the estimator found by pre-training a transformer with in-context learning (PT) behaves similarly to the optimal Bayesian estimator with prior given by the pre-training task distribution when $M$ is below a threshold, but when $M$ exceeds this threshold, PT behaves more similarly to the optimal Bayesian estimator with prior being the population distribution of tasks. Moreover, experiments show that this threshold appears to grow only linearly with data dimension $D$, and can be reduced via weight decay. Additionally, when $M$ exceeds the threshold, using more samples from the pre-training tasks during pre-training brings PT closer to the optimal Bayesian estimator with population prior, and increasing dimension $D$ does not significantly affect the behavior of PT, while it severely degrades the performance on the population task distribution of the optimal Bayesian estimator with prior being the pre-training task distribution.

**Strengths:**

1. The paper addresses the highly relevant problem of understanding the ability of large transformers to perform in-context learning, of which little is known.

2. The paper is very well written. The motivation is strong and the formulation is clear. The experiments are well-explained. I did not notice any typos.

3. The experiments are rigorous.

4. The setting is simple and easy to understand, and simultaneously the experiments reveal a number of surprising phenomenon relevant to understanding in-context learning. Thus, this paper should open doors to a number of important follow-up theoretical analyses.

**Weaknesses:**

1. A downside to the simple formulation -- it is not clear whether the observations here generalize to the large language models and textual data used in practice, which the authors acknowledge. Nevertheless, we must understand this simple setting before generalizing to more complex scenarios.

2. The paper does not provide any theoretical analysis even though the setting is ripe with open theoretical questions and possibly tractable answers. However, the experimental contributions alone are novel and significant enough to merit acceptance in my opinion.

**Questions:**

N/A

**Limitations:**

Yes

---

> ### Author Rebuttal · Authors · 2023-08-10
>
> We appreciate the reviewer’s positive impression of our paper and hope they will champion it.
>
> We agree with the reviewer’s comments that while this formulation is simple and the connection to language modeling is less clear,  understanding the fundamental role of pretraining data/task diversity in this simpler setting will provide important intuition for more complex problems.
>
> Along these lines, we note intriguingly,  an interesting recent paper that attempts to develop a data diversity metric for language data: “Beyond Scale: the Diversity Coefficient as a Data Quality Metric Demonstrates LLMs are Pre-trained on Formally Diverse Data” https://arxiv.org/abs//2306.13840.  We became aware of this paper after our submission.  This paper is along the lines of what the reviewer is asking for: a connection between dataset diversity and performance within the context of language and LLMs.  It remains for further exploration the relationship between our results on task diversity and its impact on in-context learning performance on new regression tasks, and the new paper’s data diversity metric and its ability to predict in-context learning performance on new language tasks.  However, exciting future work will be to explore the role of data diversity and its impact on transformer learning in many domains ranging from the relatively theoretically tractable settings of regression in which we know how optimal solutions behave, to the relatively open ended domains of language, vision, and multimodal data, in which new measures of dataset diversity that are predictive of performance may need to be derived.  For example, our work may motivate exploring parametric variations in the data diversity coefficient of datasets used to pretrain transformer LLMs to see if their learning performance exhibits a phase transition in data diversity - this would allow us to find perhaps small datasets of minimal diversity just above the threshold required to solve many language tasks.  We will add these ideas to our discussion section of the camera ready version.
>
> Finally, since we work with multi-layer transformers and softmax self-attention, a direct theoretical analysis in this exact setting is challenging. However, we aimed to firmly and rigorously establish the existence of the task diversity threshold empirically and understand how it behaves as we change dimensionality of the problem (paper Fig 5), regularization (paper Fig 6, weight decay) and capacity (paper Fig 6, embedding and new experiments in rebuttal). We hope this will serve as a strong starting point for future theoretical studies.

---

> > ### Comment · Reviewer_CEjq · 2023-08-18
> >
> > Thank you to the authors for your detailed response, and for your suggestion of the insightful reference. I also appreciate the authors' and other reviewers' thorough discussions. Having read these, I have decided to maintain my positive score. Nevertheless I would encourage the authors to add the new experiments and discussion, especially as it relates to whether the task diversity affect is an artifact of undertraining, to the updated paper.

---

### Official Review · Reviewer_Uwh7 · 2023-06-18

**Soundness:** 3 good
**Presentation:** 4 excellent
**Contribution:** 3 good
**Rating:** 7
**Confidence:** 4

**Summary:**

The paper studies the generalization abilities of a next token prediction pre-training with Transformers architecture to solve new unseen tasks. It performs a series of controlled experiments on simple multivariate regression tasks. The authors show that, unlike suggested in previous works, the solution learned by the transformer architecture deviates from that of the Bayesian solution with finite-support prior over tasks (dMMSE) and converges to the Ridge solution in the finite-set regime past a certain threshold. They empirically characterise when this transition occurs and study how different factors influence it.

**Strengths:**

1. The paper is well-written and introduces all the necessary background needed for understanding.
2. The paper studies the generalization of transformers pre-trained with next token prediction to solving novel tasks in-context in a few-shot manner which is an important direction to understand how models such as GPT generalize to solve language tasks seemingly different from what they were trained on.
3. The empirical study is well controlled and supports the conclusions made.


**Weaknesses:**

1. While described by a new term, “in-context learning” (ICL), the setting studied in the paper is practically identical to that of few-shot and meta-learning. Therefore, it is important to position w.r.t. this line of work, including the studies on the role diversification of pre-training tasks, e.g., [1-3]. See also [4] for theoretical foundations on learning to learn.
2. The problem of underfitting is not studied sufficiently well. It is mentioned in the paper that, in the main experiment (Fig. 1), the PT model underfits as the error on the training tasks is higher than that of the optimal solution. This would be an important aspect to clarify as one would expect that a model with sufficient capacity should converge to the optimal solution on the training dataset, i.e., dMMSE.

[1] M. Yin, G. Tucker, M. Zhou, S. Levine, and C. Finn, “Meta-Learning without Memorization.” arXiv, Apr. 27, 2020. doi: 10.48550/arXiv.1912.03820.

[2] R. Kumar, T. Deleu, and Y. Bengio, “The Effect of Diversity in Meta-Learning.” arXiv, Nov. 24, 2022. doi: 10.48550/arXiv.2201.11775.

[3] K. Hsu, S. Levine, and C. Finn, “Unsupervised Learning via Meta-Learning.” arXiv, Mar. 21, 2019. doi: 10.48550/arXiv.1810.02334.

[4] J. Baxter, “Theoretical Models of Learning to Learn,” in Learning to Learn, S. Thrun and L. Pratt, Eds., Boston, MA: Springer US, 1998, pp. 71–94. doi: 10.1007/978-1-4615-5529-2_4.


**Questions:**

1. (See point in weaknesses for more context). Is the underfitting seen in Fig.1 top left due to the evaluation on the new data points (for the same tasks) or insufficient model capacity? If latter, did authors try to increase capacity with parameters other than the dimensionality of the embedding in Sec 3.3, e.g., the depth/number of attention heads? It would be interesting to see whether there is an implicit bias in the transformer architecture (trained with SGD) regardless of its capacity.
    - Note that this observation also contradictions to the statement made in L124: "A PT that minimizes the pretraining loss LTPretrain will behave like this estimator."
2. What are “fundamentally” new tasks? This term is mentioned multiple times, but it is not defined what makes some tasks “fundamentally” new. Moreover, the new test tasks are sampled from the same distribution as the training ones ($w_i \sim N(0, I)$). See the next question for a broader discussion.
3. What makes the observed behavior (the transition from dMMSE -> Ridge) emergent and different from a superior iid generalization of one algorithm (next token prediction with Transformers) over the other (Bayesian dMMSE)?
    - In standard supervised learning, iid generalization occurs from training to test examples sampled from the same distribution. In this case, one learning algorithm, e.g., ResNet trained with SGD, can exhibit better iid generalization than another, e.g., AlexNet, i.e., converges to the true solution with fewer training examples.
    - Similarly, the paper shows that one (meta-)learning algorithm (next token prediction with Transformers) is superior to the Bayesian dMMSE algorithm.
    - More specifically, the setting of ICL can be described in terms of standard supervised learning (perhaps, with varying dimensionality of input). Indeed, the input can be defined as $x = S_k$ and the output as $y_k$, then training examples are $(S_k, y_k)$.
    - Are these analogies (fairly) the same, or is there a fundamental difference?


**Limitations:**

1. Similar to the raised question about the similarity to a standard iid generalization setting, the experimental setup of this paper does not consider the distribution shift occurring between train and test tasks, an important aspect of generalization (most probably) occurring between LM pre-training and natural language tasks.
    -  Sec 5 of the paper mentions that the diversity of the pre-training dataset of tasks is important. However, this is only supported when more tasks are sampled from the same distribution as test tasks ($w \sim N(0, I)$ in both train and test). Whereas in language, there is (most probably) a distribution shift between them.
2. The paper limits the study to transformer architectures. It would be interesting to see how other architectures used in few-shot and meta-learning behaves in the same settings, e.g., [1]

[1] M. Garnelo et al., “Conditional Neural Processes.” arXiv, Jul. 04, 2018. doi: 10.48550/arXiv.1807.01613.

---

> ### Author Rebuttal · Authors · 2023-08-10
>
> **Weaknesses:**
> - We agree it’s important to add a discussion positioning our work with respect to the few-shot and meta-learning settings, particularly considering the role of task diversity. Thank you for the references; we will add them to our paper and discuss them and others.
> - Please refer to the main rebuttal response for a discussion on underfitting (also as mentioned in Question 1).
>
> **Effect of model capacity (Question 1):** We further investigate the role of capacity in our Small PT. We hypothesize that a smaller model will be in a more capacity-constrained regime, so that ablations of key parameters affecting capacity will have a more pronounced effect on the task diversity threshold. Our Small PT has embedding size 64, 4 layers, and 2 heads. If we start from Small PT and either increase the number of layers (Figure 1B), or increase embedding size (Figure 1C), the task diversity threshold goes up. This presents evidence that model capacity can be a key feature of this transition at small model scales.
>
> With “A PT that minimizes the pretraining loss $L^{T_\text{Pretrain}}$ will behave like this estimator,” we mean that a model that is able to attain (close to) the lowest possible $L^{T_\text{Pretrain}}$ over all functions will behave like the dMMSE estimator. For large enough $M$, a model _trained to minimize_ $L^{T_\text{Pretrain}}$ in fact does not find this minimum and so does not behave like dMMSE; we will word this more clearly in our paper.
>
> **Defining fundamentally new tasks (Question 2):** We first clarify that new test tasks used to compute $L^{T_\text{True}}$ are sampled from a different distribution than pretraining ones: to generate a pretraining sequence, we sample a task from the _uniform_ distribution over pre-selected pretraining tasks $\{w_1,\ldots, w_M\}$, whereas when we calculate $L^{T_\text{True}}$, we average over the underlying task distribution $\mathcal{N}(0, I)$.
>
> In this setup we can make the notion of “fundamentally new” precise because we have a well-defined metric on task space: tasks are regression vectors and distance is simply cosine distance in this space. We show that distance of a new in-context task from a pre-training task strongly determines performance below the task diversity threshold but not above it. This is shown in Figure 4 (main submission) when we interpolate between pre-training tasks and measure in-context learning performance.
>
> **Distribution shift (Limitation 1):** It is the case that the pretraining tasks $\{w_1,\ldots,w_M\}$ are sampled from the same _underlying_ Gaussian distribution as test tasks. It is likely that this property does not hold in language modeling settings where the pretraining tasks and test tasks themselves come from different distributions. We will discuss this further as a limitation in our paper. However, we think that understanding the simpler setting where the distributions are the same is an important first step. Searching for settings in which there is a shift in the underlying distribution and near-optimal generalization persists is an exciting direction for future work.
>
> **Q3: Connection to supervised learning and IID generalization:**
> Indeed, this problem can be framed as an instance of supervised learning where at each context, $k$, the datapoint is $S_k$ and the target is $y_k$. Standard IID generalization in our setting, however, corresponds to strong performance on $L^{T_\text{Pretrain}}$, for which test sequences are distributed identically to pretraining ones, with tasks sampled from a finite pool of $M$ regression vectors. In fact, dMMSE is the optimal model for _IID_ generalization. Conversely, good performance on $L^{T_\text{True}}$ means that the PT is generalizing _out of distribution_ in that it is able to perform well on _new_ tasks drawn from $\mathcal{N}(0, 1)$ it was not trained on. The key is that, even though the set of pretraining tasks were sampled from the Gaussian, once those are fixed and we start pretraining, the distribution of $(S_k, y_k)$ is different from the distribution of $(S_k, y_k)$ used to calculate $L^{T_\text{True}}$. What makes this behavior emergent is precisely that even though the model is trained to minimize $L^{T_\text{Pretrain}}$, it ends up performing well on $L^{T_\text{True}}$, and at the expense of not minimizing $L^{T_\text{Pretrain}}$ (see Figure 2, top left in main text). Moreover training with *more* data per task makes the model more like Ridge at high task diversity and more like dMMSE at low task diversity.  This asymptotic approach to two *different* task-diversity dependent estimators given more training data is a distinct phenomenon that does not show up in classic learning theory (see our new results in Rebuttal Fig. 2AB and global rebuttal).

---

> > ### Comment · Reviewer_Uwh7 · 2023-08-12
> >
> > Thank you for the clarifications and additional results. One thing that still concerns me is the argumentation about the shown behavior to be an "emergent" one. It still seems a bit misleading to me. As also noted by vcDA, this might be simply the inability of the model to fit the optimal dMMSE solution for large $M$s due to the limited capacity. This is also in line with the new results provided in Fig.1(B), where larger models can fit dMMSE solution for larger $M$s => an increased task diversity threshold.
> >
> > At the same time, I believe that the experimental setting considered in this work and the results provided through rigorous experimentation would be a valuable contribution to the field, regardless of their current and future interpretation. For example, even if it is a "simple underfitting," the fact that the model converges to the Ridge solution and not any other solution could be an interesting insight. I suggest that authors use "emergence" more carefully and discuss other alternative explanations clearly and explicitly. I am reconsidering my final grade, and I will follow the further discussion for that.

---

> > > ### Author Response · Authors · 2023-08-15
> > >
> > > Thank you for the follow-up comments! We’ll be very clear in our paper about how we define emergence, which is that, when the number of tasks is above the task diversity threshold,  although the PT model is trained on samples from $T_{\text{Pretrain}}$:
> > > 1. The PT achieves better performance on samples from $T_{\text{True}}$ than the Bayes-optimal estimator with prior $T_{\text{Pretrain}}$.
> > > 2. Moreover, above and beyond (1), the PT achieves strong (near-optimal) performance on samples from $T_{\text{True}}$, despite the fact that $T_{\text{True}}$ and $T_{\text{Pretrain}}$ are different.
> > >
> > > We certainly agree that it’s important to use this term carefully, that is adhering to a definition while also explaining possible causes, including the possibility that either the model is capacity-constrained and not expressive enough to implement the dMMSE solution, or the model is expressive enough to express dMMSE but this solution has such a small basin of attraction that SGD cannot find it.
> > >
> > > However, regardless of why SGD for the PT cannot find the dMMSE solution when $M$ is bigger than the task diversity threshold, **we believe that a fundamental finding of our paper is that, out of the infinitely many in-distribution underperforming solutions for $T_{\text{Pretrain}}$ that the PT could find, it remarkably finds the Ridge solution, which is optimal for the out-of-training distribution $T_{\text{True}}$**. It is precisely this behavior that we are calling emergent (as defined in joint points 1 and 2 above).  This behavior is surprising and emergent precisely because nothing in $T_{\text{Pretrain}}$ necessarily pre-ordained this. We will be very careful to elucidate that emergence in this context means this and nothing more.
> > >
> > > Finally, a way to think about the dependence of the task diversity threshold on model capacity is as follows: think about a 2D phase diagram where one axis is task diversity (i.e. $M$) and the other axis is model capacity ($C$).  At each point in the 2D phase diagram, we can ask a question: if we train a model with capacity $C$ and task diversity $M$ on more data (either more iterations as in Rebuttal Figs. 1A and 2A right and paper Fig. 3, or larger batch size as in Rebuttal Fig. 1A and paper Fig. 2) then, does the PT become more like Ridge or less like Ridge (and closer to dMMSE)? The answer to this binary question yields a phase transition boundary in the 2D phase diagram where on one side of the boundary (i.e. small $M$ and large $C$) the PT becomes more like dMMSE, and on the other side of the phase boundary, (i.e. large $M$ and small $C$) the PT becomes more like Ridge.  We have been implicitly delineating this phase boundary by finely sampling $M$ and coarsely sampling $C$. It is entirely expected that the phase boundary should depend jointly on $M$ and $C$, hence the task diversity threshold in $M$ should grow modestly at least with $C$. We do not view this as a negative result of our work, but rather the elucidation of an important phase transition in a two parameter joint space of task diversity and model capacity.

---

> > > > ### Comment · Reviewer_Uwh7 · 2023-08-17
> > > >
> > > > I thank the authors for the provided discussion, which clarifies my questions. I generally agree with the concerns raised by vcDA. However, I believe the contribution can still be valuable enough for the community and increase my grade to 7, advocating for acceptance (provided the authors include more explicit discussion about the emergence and under-parametrization in the camera-ready version).

---

### Official Review · Reviewer_NkJG · 2023-06-30

**Soundness:** 4 excellent
**Presentation:** 4 excellent
**Contribution:** 2 fair
**Rating:** 7
**Confidence:** 4

**Summary:**

The paper investigates in-context learning in the restricted setting of linear regression. Diverging from recent work, they focus on studying the  effect of task diversity i.e. the number of different linear teachers used during pre-training of Transformer models.
Surprisingly, they find that although fitting the training data optimally, the Transformer outperforms a Bayesian estimator in an intermediate stage of diversity coming closer to the optimal solution of ridge regression. This implies an implicit regularization in the pre-training of the Transformer favoring the better generalizing solution.

**Strengths:**

The paper studies a highly relevant problem, that of in-context learning, in a restricted setting of linear regression. I find the paper very well written, easy to follow and insightful. Well done!



**Weaknesses:**

The major concern about this paper is that it provides only minor novel insights in my opinion. We know that Transformers when pre-trained on enough data implement functions that resemble gradient descent and act as the ridge regression solution. Now given a fixed amount of
teacher weights, if the Transformer is large enough simply learn and remember all needed weights ("features") and implement dMMSE.
Now, when the capacity is not sufficient anymore it falls back to ridge regression which happens to generalize better.
In my opinion, this just confirms classic statistical learning theory, here with the nice twist of Transformers implementing algorithms on both extremes. Please correct me if I am wrong here.
Although I find the paper still interesting, for me to give a higher score various questions and experiments are needed.

I have a couple of open ended questions on which I would like to hear your thoughts. See questions.

Minor things:
One detail that confused me a bit where the different start and end points of the x-axes in the plots. Maybe highlight that in the text or visually? It makes sense to separate it as you did.



**Questions:**

Unfortunately, to give a higher score I feel that the paper should go in a slightly different direction.
Questions for a higher score:

Which architecture modifications lead to faster/slower generalization? #Heads, Layers? Do these modifications scale linearly with the number of teachers or is another scaling observed? It seems that dMMSE requires non-linearity possibly provided by the softmax in the attention layer. What happens if you train linear self-attention Transformers on your settings. Do you generalize faster?


Open ended:
1) Did you check other task diversities? It could be interesting to investigate e.g. different "modalities"  i.e. where the teachers are sampled from vastly K different curvatures? Maybe there are other distribution changes that make sense.
The goal would be to understand if one can push towards the generalizing solution with the same amount of data.

2) Follow up on 1): Regarding your hypotheses of implicit regularization. It would be very interesting imo to check different initializations that usually are key to change between lazy and feature learning. It would be great to see if that intuition checks out or if there are other forces
at play here.

3) Can you plot the first two attention heads of the layer 1 and 2 (maybe more) for me for a model that is before in the middle and after the phase transition. My claim (maybe obvious) is that there will be visible differences here and that solutions that are close to ridge regression show copying behavior in the first layer to group together to do GD (see e.g. https://transformer-circuits.pub/2022/in-context-learning-and-induction-heads/index.html, https://arxiv.org/pdf/2212.07677.pdf) which might not be necessary to be implemented for dMMSE.

**Limitations:**

Limitations are obvious and mentioned in the discussion.

---

> ### Author Rebuttal · Authors · 2023-08-10
>
> We thank the reviewer for their comments which we address below:
>
> **Novelty and significance of contribution (Weakness):** As the reviewer notes, it has been shown that if the pretraining distribution has infinite task diversity (each pretraining task is sampled i.i.d. from a Gaussian distribution), PTs implement Ridge regression. It is also perhaps unsurprising that if the pretraining distribution has a small number of tasks relative to the model capacity, the PT will memorize the tasks and implement dMMSE. However, we think it isn’t obvious that PTs would fall back to _Ridge regression_ if they do not have the capacity to _exactly implement dMMSE_. Specifically, consider a setting after the task diversity threshold, eg. at $2^{16}$ tasks in paper Fig. 2, where the _pretraining loss_ of the dMMSE estimator is significantly lower than that of Ridge. One natural hypothesis is that, since the PT is optimized to minimize the pretraining loss, even if it doesn’t have the capacity to exactly implement dMMSE, it will achieve a solution at an intermediate value of loss between dMMSE and Ridge. Under this hypothesis, if trained for longer, the PT would reduce its pretraining loss further, thereby becoming _less like_ Ridge ($\Delta_\text{PT,Ridge}$ would increase), until it converged to the closest solution to dMMSE that its capacity allows. However, we find that this is not the case: above the threshold the PT seems to “prefer” the Ridge solution. In our paper, and in the additional experiments in the rebuttal, we aim to establish that this “preference” is a robust feature and not just an artifact of undertraining (see the global rebuttal). To our knowledge, such a result has not been shown for transformers and we are not aware of results from classical statistical learning theory that would imply this without _assuming_ a biased function class. This finding is also significant as it shows an exciting example where, even when trained on finite task diversity (as one would expect to in practical settings), PTs show a bias for learning the underlying generative model of the tasks instead of just minimizing the pretraining loss.
>
> In particular, we encourage the reviewer to see rebuttal Fig. 2 and the global rebuttal which reveals a remarkable and sharp phase transition in learning dynamics. To summarize: at small task diversity $M$, pre-training learning curves initially go towards ridge but then turn away and go to dMMSE. However, as $M$ increases, around the task diversity threshold, *suddenly*, learning curves escape dMMSE and move towards ridge *no matter how long you train despite the fact that the pre-training loss global minimum is dMMSE*. Rebuttal Fig 2B shows that this transition would not be predicted by a simple scaling analysis of early stopping time. We are unaware of classic results in statistical learning theory that can explain all these facts. We provide a conjectured loss landscape explanation in our global rebuttal.
>
> **Questions**:
>
> **Architecture modifications:** We further investigate architecture modifications and their effect on the transition in our Small PT. We hypothesize that a smaller model will be in a more capacity-constrained regime, so that the effects of architectural parameters on the task diversity threshold will be larger. Switching from Base PT to Small PT, which has embedding size 64, 4 layers, and 2 heads, leads to a task diversity threshold 8x lower. Furthermore, starting from Small PT, if we increase the number of layers (cf. rebuttal Fig. 1B), or embedding size (cf. rebuttal Fig. 1C), the task diversity threshold goes up. This shows that restricting the models class does in-fact reduce the task diversity threshold. Unfortunately, this also reduces the overall performance of the models, revealing a tradeoff between model performance and required diversity of dataset.
>
> **LSA:** The softmax might in fact play an important role in allowing the model to reach dMMSE-like solutions. We experimented with linear self-attention (LSA) and found training to be unstable, even after some hyperparameter tuning. A key difference that might cause this is that other works considering LSA only predict the final test example while we predict at each context length. Given that this isn't a simple modification, we think that investigating LSA is beyond the scope of this paper but an excellent opportunity for future work.
>
> **Open ended**:
>
> **Other task diversities.** We performed experiments where pretraining tasks are sampled from a Laplace prior, instead of a Gaussian (see Figure 3). In this setting we again find a task diversity threshold. Crucially, for $M>2^{12}$, the PT achieves a lower test error on new tasks drawn from the Laplace prior than does ridge regression (~0.34). This suggests that the PT is primed to generalize to unseen tasks from the underlying distribution that the pretraining tasks are sampled from. Thus, it might be tricky to engineer a different prior to sample tasks from to enable faster generalization. But this is a very promising direction for future work where the tasks come from hierarchical or otherwise structured priors that we might be able to exploit.
>
> **Lazy vs. feature learning:** It would indeed be interesting to use our setup as a tool to not just study implicit bias in this problem but also for getting a better understanding of lazy versus feature learning.
>
> **Evidence for copying and GD behavior in attention layers:** We have not had the opportunity this week to plot early attention layers. It’s certainly plausible that there will be noticeable differences for dMMSE vs. Ridge-like models, especially if, as the reviewer suggests, the PT is implementing a version of copy + GD in the post-transition regime. As alluded to in the LSA discussion, however, we’re not sure the solution would appear structurally as simple in our setting of large models with MLP’s and softmax attention, as well as predicting at every context length.

---

> > ### Comment · Reviewer_NkJG · 2023-08-10
> > **Thank you!**
> >
> > I have read the rebuttal, the other opinions and will increase my score, voting for acceptance. Thank you!

---

### Official Review · Reviewer_nyxK · 2023-07-05

**Soundness:** 3 good
**Presentation:** 3 good
**Contribution:** 2 fair
**Rating:** 6
**Confidence:** 4

**Summary:**

The paper analyzes in-context learning capability of auto-regressive (continuous) Transformer models where the training data consists of linear regression problems. Paper finds that the ability to ICL with optimal solutions for the given prior (e.g. ridge regression for noisy linear regression) emerges only after the model is exposed to a certain number of different regression tasks (or simply a certain number of $w$s) from the prior distribution. After this threshold has been passed, however many instances they provided, the model choses Ridge regression instead of local Bayes optimal solution for the particular training dataset (called dMMSE).

**Strengths:**

1. Finds a threshold for task diversity that makes the Transformer model switch from local bayesian solution (dMMSE) to optimal bayesian solution (Ridge regression).

2. Find that weight norm improves this diversity threshold in the particular setup.

3. A clear and concise writing.


**Weaknesses:**

1. The results show this threshold exists even if they increase the number of instances per task, and number of training iterations. We do not know if the same transitions hold if the authors increase the number of instances per task and number of training iterations at the same time? Can you try an experiment where you maxed out everything: model size, training iterations and number of instances per task, and verify the results?

2. If 1 holds, what is the justification or intuition for this transition? As far as I saw, there was no discussion of possible theories to explain this behaviour.

3. The experiments are only in linear regression with Gaussian priors, and previous work (e.g. Garg et al. 2022) had setups with sparse regression and non-linear problems.

4. Would be more complete if analysis of hyper parameters can be extended to the number of layers too.






**Questions:**

1. Are these transitions hold for different distributions than Guassian (Laplace, etc)?

2. L75-79: Could you simplify the message here by giving the scaling laws (linear, logarithmic, or by its shape) of threshold given the task dimension. You can get this by running with more dimensions, for example, 2 to 128 dimensions and plotting the point where $\Delta$dMMSE crosses $\Delta$PT.  We can do same for weight decay and embedding experiments too. In my opinion, this type of plot would be much more explanatory than what Figure 5 and 6 currently has. And I do not think the different batch sizes are needed in these plots, why do not pick the highest one?

3. Fig 6: Do we need 4 plots for different values of weight decay/embedding? Could you plot them at the same figure? Same for Fig5 left three panels. Also see my suggestions above.

**Paper Summary**
I think this paper finds something surprising that the models transition from dataset specific optimal solution to global prior optimal solution. But the results needs some verification and explanation as I queried in above.


**Limitations:**

The authors could discuss the possibility of this transition might not be hapenning in the case of enough iteration and samples per task.

---

> ### Author Rebuttal · Authors · 2023-08-10
>
> **Undertraining (Weakness 1 + Paper Summary):** To clarify, there are two independent parameters in our experimental setting that control how we scale the amount of data the model is trained on. We can vary the number of pretraining tasks, $M$, which is the x-axis in our experiments. We can also vary the total number of examples seen during training by increasing the batch size or training duration. The number of examples per task is the total number of examples seen divided by $M$. So for any fixed $M$, increasing the number of iterations increases the number of instances per task at the same time.
>
> However, there is still a question of if the task diversity threshold is caused by undertraining, which is what we think the reviewer’s question is about. It is possible that for a model to converge to dMMSE, when we increase $M$ by a factor of 2, we also need to increase the total number of training iterations by a factor of 2, thus keeping the total number of samples per task constant for each $M$. We discuss this question in detail with additional experiments in the global rebuttal. To summarize, we do a version of the “maxing out experiment” the reviewer asks for with a smaller model (which was necessary to finish the experiment within the week). For this Small PT, we train on 8x the amount of data (with larger batch size AND more iterations). We find that the location of the task diversity threshold is in fact robust. Since we performed a similar experiment in our paper (in Figure 2, we 4x the data when we go from batch size 256 to 1024), we believe that similar results will hold for larger models. We also provide additional evidence by looking at how the early stopping time for Ridge (the training step at which the model is closest to Ridge) scales with $M$; for details, please refer to the global rebuttal. This provides further evidence that there is a real change in the training dynamics of the models near the task diversity threshold and beyond the threshold, models really do prefer the Ridge solution no matter how long one trains them for.
>
> **Possible explanation for the transition (Weakness 2):** While we don’t like to speculate, given the reviewer’s request we provide our best current guess which is motivated by the picture of learning dynamics revealed in rebuttal Fig. 2. To summarize, we think that as the task diversity increases, there is a transition in the geometry of the loss landscape that makes the dMMSE solution too hard to find and the model instead picks the Ridge solution even though it has higher error. Please see the global rebuttal for a detailed discussion.
>
> **Exploring more hyperparameters (Weakness 4):** We further investigate the effect of capacity on this transition using our Small PT model (see global rebuttal). We hypothesize that a smaller model will be in a more capacity-constrained regime, so that ablations of model depth and embedding dimension will have a more pronounced effect on the task diversity threshold. If we start from Small PT and either increase the number of layers (rebuttal Figure 1B), or increase embedding size (rebuttal Figure 1C), the task diversity threshold goes up. This adds results for the effect of layers and presents evidence that model capacity can be a key feature of the task diversity transition at small model scales.
>
> **Laplace prior (Weakness 3 and Question 1):** We investigate if the task diversity threshold is also present when tasks are drawn from a different prior. We performed new experiments where pretraining tasks are sampled from a Laplace distribution, instead of a Gaussian (Rebuttal Fig. 3). In this setting we again observe a task diversity threshold in the model’s error where, beyond the threshold, more training leads to better performance on new, unseen Laplace tasks. We note that for $M>2^{12}$, the PT achieves a lower error on new Laplace tasks than ridge regression (~0.34). This suggests that the PT is primed to generalize on unseen tasks from the same underlying distribution that the pretraining tasks are sampled from.
>
> **Simplifying takeaways (Question 2 and 3):** We note training at very large task dimensions requires larger models to perform well and also exhibits many loss spikes which makes training unstable and very expensive. So given the number of training runs we need to conduct, experiments sweeping past $D=32$ become infeasible. For our task dimension experiments (Figure 5, main submission), we speculate that the scaling is linear, but don’t want to overclaim given that we’re sweeping task dimension over a factor of 4. However, we agree that the messages in our figures could be made clearer by presenting scaling laws for settings with sufficient datapoints. We will simplify the figures and corresponding takeaways in our paper; our latest architecture ablation experiments (rebuttal Figure 1B and C) already incorporate this suggestion.

---

> > ### Comment · Reviewer_nyxK · 2023-08-15
> > **thank you!**
> >
> > Thank you for the detailed response and extra experiments. The transition seemed to me really interesting and further experiments that clarify this holds for many data points addressed my concerns. I increased my point as a result.

---

### Official Review · Reviewer_vcDA · 2023-07-07

**Soundness:** 2 fair
**Presentation:** 3 good
**Contribution:** 2 fair
**Rating:** 4
**Confidence:** 5

**Summary:**

This paper studies whether a Transformer can in-context learn linear functions. While previous work studies a Gaussian prior over the weights, this work studies a discrete prior over the weights, resembling a less diverse pre-training task distribution. The Bayes-optimal estimator for this distribution is not Ridge Regression. This allows one to measure generalization based on whether the Transformer behaves like the Bayes-optimal solution to the pre-training distribution or the more general Ridge Regression estimator. It is seen that under a certain task diversity threshold, the model learns the Bayes-optimal solution to the pre-training distribution. Above this threshold, the model seems to learn the generalizing solution of Ridge Regression, indicating that pre-training task diversity may play an interesting role in ICL generalization.

**Strengths:**

1) The paper discusses an experimental setup that allows for the investigation of how task diversity can affect ICL performance, as well as the optimal solution.
2) The paper varies batch size and step count, and both share a similar task diversity threshold where more training gets closer to ridge regression estimator.
3) The paper assesses an optimally-tuned Smoothed dMMSE to eliminate a hypothesis for what the model implements

**Weaknesses:**

Though the paper identifies the existence of a threshold, there is not a strong study into possible reasons for why. There is a strong possibility that the threshold is a consequence of the evaluation setup rather than a feature inherent to the problem, and it is important for this paper to address these concerns since it claims that the threshold exists. I believe explaining the features of the threshold or why it exists would constitute a much larger contribution. Concrete suggestions below.
1) One natural hypothesis, as noted by the authors on lines 235-236, is that the model simply doesn't have enough capacity to memorize M tasks. However, one could train a larger (or smaller model) while controlling for all other settings and seeing whether it has a different task diversity threshold. If it does, then this paper could offer insight into the features of such a threshold rather than just showing that it exists.
2) It is possible that the threshold is a consequence of plotting the loss curves without giving more training steps to higher task count problems (the solution to 64 tasks is likely harder and takes more steps than the solution to 32 tasks). Is it possible to show the distance to Ridge over training time to rule out this possibility? Moreover, are there reasonable normalizations (alluded to in Question 2)?

**Questions:**

1) Is it the case that the model's distance to Ridge monotonically decreases over training or does the distance to Ridge increase and decrease? This can have important implications for the conclusion and determining whether the models for higher task count simply need to be trained longer, since it is possible the exact step count presented is still in the process of getting closer to Ridge and would later go further away.
2) For the models that converged to sMMSE, how many steps does it take? Is it the same number of steps across task counts?
3) (Minor) In Figure 2, is the upper right figure the same as the lower right figure?


**Limitations:**

The authors do not have a separate limitations section but discuss alternative explanations for experiments throughout the paper.

---

> ### Author Rebuttal · Authors · 2023-08-10
>
> We thank the reviewer for their insightful questions which motivated new experiments which we hope are compelling:
>
> **Effect of model capacity on task diversity threshold (Weakness 1):** In the rebuttal, we further explore the effect of model capacity on the task diversity threshold. We use a Small PT with 64 dim embeddings, 4 layers and 2 attn heads. We hypothesize that this model is more capacity constrained than the base model in our paper (Base PT) and so ablating parameters affecting capacity will have a larger effect on the task diversity threshold. We observe that changing from Base PT to Small PT leads to a drop in the task diversity threshold by a factor of 8 from $2^{14.5}$ to $2^{11.5}$. Furthermore, if we start from Small PT and either increase the number of layers (Fig 1B), or embedding size (Fig 1C), the task diversity threshold goes up, demonstrating that, when starting with smaller models, model capacity can be a key feature of the task diversity transition. In our paper, we showed task dimension (Figure 5 main text) and weight decay (Figure 6 main text, top) could also control the task diversity threshold. We believe that training Base PT with weight decay decreases the effective capacity of the model, hence lowering the task diversity threshold. Together these results provide a more extensive characterization of how layers, embedding dimension, weight decay, and task dimension all affect the task diversity threshold, as requested by the reviewer.
>
> **Is the task diversity threshold an artifact of undertraining (Weakness 2, Question 1 + 2)?** We refer the reviewer to the discussion of undertraining in the global rebuttal. Here, we emphasize a few key points. Following weakness 2 + question 1, we plot distance to Ridge training curves for 500K and 2M steps (rebuttal Fig. 2A). At either training duration, training curves fall into 1 of 3 groups: (1) at very low task diversity (very small $M$, not shown in figure) models monotonically move further from Ridge, (2) at low task diversity (small $M$) models move closer to Ridge then move away, finding the dMMSE solution, (3) at high task diversity (large $M$) models monotonically move closer to Ridge. The transition from pattern (2) to pattern (3) happens near the task diversity threshold. Comparing Fig 2A left and right panels, we note that if one increases the training time even **by a factor of 4** from 500K to 2M steps, this dichotomy between the behaviors of training curves on either side of the task diversity threshold is preserved. Also, form Question 2 about normalization of training steps, it is indeed harder to learn dMMSE for larger values of $M$: it takes longer for a model trained on larger $M$ to stop moving towards Ridge and turn around and move towards dMMSE. However, we note that the hypothesis of undertraining generated from this observation is unlikely to explain the task diversity threshold given that we train for **much longer** than the longest time it takes for models in group (2) to turn away from Ridge. Concretely, the green curve for $M=2^{10}$ in Fig 2A right turns around at $t^*$ ~ 100K steps. The red curve for $M=2^{11}$ increase $M$ by a factor of 2 and yet, continues to move towards Ridge even though it is trained for 2M steps, more than 20x the $t^*$ noted for $M=2^{10}$. Thus the curves at high task diversity whose distance to ridge monotonically decreases are unlikely to suddenly turn around and depart from ridge after even longer training. To bolster this picture quantitatively, we perform a scaling analysis of the time $t^*$ it takes for curves in group (2) i.e at small M to turn around and start to move further from ridge with more training. This analysis reveals that this time $t^*$ scales roughly as $\sqrt{M}$ (rebuttal Fig. 2B). However there is a sudden phase transition around the task diversity threshold where the distance to ridge curves start decreasing monotonically and don’t turn around even when trained for times **much** longer than the $\sqrt{M}$ turn around time extrapolated from below the task diversity threshold (sudden large jump in the points in rebuttal Fig. 2B). For example, in rebuttal Fig. 2B, at $M=2^{11}$ task diversity, the $\sqrt{M}$ scaling would predict the a turn around time around $t^*=10^5$. However, the training curve *doesn't* turn around for atleast 2 million training steps instead becoming monotonically closer to ridge.  This sudden and dramatically large change in behavior of the training curves between $M=2^{10}$ and $M=2^{11}$ vividly demonstrates the fundamental nature of the task diversity threshold phase transition and how it is highly unlikely to be an artifact of not enough training time at large $M$; for example training even longer at $M=2^{11}$ is unlikely to yield turn around behavior, and if it did, it would violate the $\sqrt{M}$ scaling law in rebuttal Fig. 2B.  Finally, as further evidence (rebuttal Fig. 1)  we not only increased the amount of training data by 2x as in most of our experiments, **but also increased it by 8x** compared to our base setting: this did not change the task diversity threshold.
>
> Taken together these results provide strong evidence that our discovered task diversity threshold is not an artifactual consequence of our evaluation setup but rather it is inherent to the learning dynamics of the transformer.
>
> The reviewer asked why there is a task diversity threshold phase transition. While we don’t like to speculate, given the reviewer’s request we provide our best current guess in the global rebuttal, motivated by the  picture of the learning dynamics revealed  in Rebuttal Fig. 2. Please see global rebuttal for more details.
>
> **Question 3:** No. Fig. 2, upper row: evaluated on finite set of pretraining tasks, lower row: evaluated on new tasks from Gaussian dist.

---

> > ### Comment · Reviewer_vcDA · 2023-08-12
> >
> > Thank you, I appreciate the new experiments! I find the order of learning Ridge than dMMSE important and I hope it gets adequately situated within the discussion of the threshold. I am currently reconsidering my score, and I wanted to ask some questions before I do so.
> >
> > The results suggest that the task diversity threshold for Small PT is $2^{11.5}$ and the task diversity threshold for PT is approximately $2^{14.5}$. From this, I'd imagine that as you increase model capacity, the task diversity threshold increases. I believe for large $M$, the models are not in an over-parameterized regime [1] since the model can not solve the training distribution (evidenced by not achieving dMMSE performance).
> >
> > **Candidate Explanation**: The "task diversity threshold" is the model only having the capacity to represent Ridge and dMMSE for task counts below the threshold and the model not being able to express dMMSE for task counts above the threshold. As for the improved out-of-distribution generalization, this is complicated by the model learning Ridge first and then dMMSE for medium-sized $M$. As such, the model learning Ridge does not reflect out-of-distribution generalization as much as it reflects the model not generalizing in-distribution (due to capacity constraints).
> >
> > I was curious whether the experiments shed any insight into whether this explanation is feasible. It would help me understand how this work expands our current understanding of generalization in ML, especially with respect to the standard bias-variance tradeoff (or the under-parameterized regime of double descent).
> >
> > [1] Reconciling modern machine learning practice and the bias-variance trade-off

---

> > > ### Author Response · Authors · 2023-08-15
> > >
> > > We are glad you appreciate the new experiments, and we thank you for your important questions, which we attempt to answer. The reviewer correctly notes that as model capacity (at least as measured in number of parameters) increases, so does the task diversity threshold $M^*$, especially when the model capacity is small to begin with.  Now when $M > M^*$ for any given model capacity, our simulations indicate that our model does not find the dMMSE solution which would minimize in-distribution error $L^{T_\text{Pretrain}}$.  There could be two main reasons for this: (1) there is a model capacity/expressivity issue, namely there is *no* configuration of model weights that can achieve the dMMSE estimator when $M>M^*$, or (2) there is a trainability issue, namely, there *is* a configuration of model weights that could achieve dMMSE, but its basin of attraction under SGD is so small that SGD starting from a random initial condition does not find it, and instead SGD typically is driven to the training error minimum of a basin containing a *different* solution. The reviewer suggests (1) as a possible explanation.  Our current experiments may be consistent with explanation (1), but also do not rule out explanation (2) as a possibility. To rule out explanation (2) one would have to find the dMMSE solution without SGD, and it is not clear how to do that. This is characteristic of all failures of machine learning: when it fails we do not know if the failure is due to expressivity alone or due to trainability via SGD.
> > >
> > > That being said, even in a setting in which the model cannot fit dMMSE (regardless of whether the failure is due to expressivity or trainability), there exists an infinite set of underfitting (in-distribution) solutions which achieve lower in-distribution training error. Apriori, it is unlikely that, at a given threshold value of $M$, all of these underfitting solutions become inaccessible.  However, our experiments show that: when the tasks are drawn from a Gaussian, beyond the task diversity threshold, amongst the infinite set of all possible underfitting solutions, _there is a preference for ridge regression as **the** underfitting solution_, and the PT reaches $L^{T_\text{True}}$ performance on par with models trained in the infinite task setting.  When the tasks are drawn from a Laplace distribution, in turn, beyond the task diversity threshold, there is a preference for the Bayes-optimal MMSE estimator under a Laplace prior. **Hence, the model doesn’t just learn any underfitting solution. It learns the solution that optimizes error for the underlying generative model of pretraining tasks. The fact that it performs on par with the infinite task setting solution, despite having access only to a finite set of tasks, means that it is, remarkably, correctly generalizing out of the training distribution, even though it is not generalizing in-distribution.** This an interesting example of how the failure to generalize to a limited diversity in-distribution setting does not mean one cannot generalize to a more important, diverse out-of-distribution setting. This setting is not often considered in classical learning theory.
> > >
> > > Finally, we note that the reviewer’s description of the model’s “order of learning Ridge than dMMSE” is indeed important; in our revision we will be giving our rebuttal Fig. 2 strong prominence in the main paper.  However, we would like to make this statement more precise. Fig. 2 does show that when $M$ is below the task diversity threshold, as measured by $\Delta_\text{PT, Ridge}$ the model does get closer to ridge over the course of training, and then moves farther away towards dMMSE. However, at these intermediate values of $M$ the lowest value of $\Delta_\text{PT, Ridge}$ achieved by the PT over the course of training is still considerably higher than that achieved by the PT in the infinite task setting. This means the PT _does not_ completely learn Ridge before starting to overfit at these intermediate values of $M$.

---

> > > > ### Comment · Reviewer_vcDA · 2023-08-15
> > > >
> > > > Thank you for this further clarification. In particular, I agree with the learning-theoretic observation as provided in the rebuttal. I did not get this message from the original paper; I believe the paper would benefit from this nuanced discussion of how SGD + Transformers are incapable of fitting the optimal in-distribution performance and it is interesting that of all possible under-fitting solutions, the model converges to the one that generalizes out-of-distribution. This new message of the work should also be situated within the context of Simplicity Bias literature, which discusses how among multiple possible solutions, models tend to converge to the simplest one, and it seems that Ridge is a simpler solution in some sense since it's discovered earlier in training and able to be learnt by smaller Transformers.
> > > >
> > > > Due to this extended discussion/clarifications about the results of the paper and the insightful clarifications provided by the reviewers, I am raising my score from a 3 to a 4. However, I maintain that showing the model converges to the Bayes-optimal solution for high task counts is a narrow contribution specific to this setup and is reliant on the 1) simplicity bias of getting closer to ridge first and 2) being in the under-parameterized regime.

---

### Author Rebuttal · Authors · 2023-08-10

We thank the reviewers for their insightful comments.
Summary of new results in the attached figures:
- We study a Small PT with 64 dim. embeddings, 4 layers, and 2 attn. heads; the smaller scale enabled us to answer questions within the week. In Fig 1A, we reproduce our key finding, the existence of a task diversity threshold.
- A Small PT, with a lower capacity, has a lower task diversity threshold than the Base PT in our paper. Since PTs at this scale are more sensitive to capacity constraints, we vary depth and embedding size on Small PTs showing that as capacity increases, so does the task diversity threshold. The results (Fig 1 B,C) are discussed in individual responses.
- Multiple reviewers asked if the task diversity threshold may be an artifact of undertraining. We train Small PT for much longer and address this question below; it is not the case.
- We investigate the effect of changing the task distribution from Gaussian to Laplace. In Fig 3, we show that a task diversity threshold also exists in this setting. We discuss this further in the response to reviewer nyxK who raised this question.

**Is the task diversity threshold an artifact of undertraining?** In rebuttal Fig. 2A, we see that for some pretraining task diversities, $M$, below the task diversity threshold, $M^*$, $\Delta_\text{PT,Ridge}$ decreases early in training until it reaches a minimum at time step $t^*$, and then increases as the PT converges to dMMSE ($M=2^8,2^9,2^{10}$ in Fig. 2A). Call $t^*$ the early stopping time for Ridge. A reviewer noted that a possible reason for the task diversity threshold could be that for $M > M^*$, $t^*$ is larger than the total training time, and so $\Delta_\text{PT,Ridge}$ appears to decrease with more training. Indeed, in an ideal experiment, we would keep the number of examples per task fixed as we increase $M$, thus training on more data for larger $M$. Unfortunately, this is infeasible for the range of $M$ we study. However, we argue that the task diversity threshold is a feature inherent to the problem and not merely an artifact of undertraining and present the following evidence:
1. For each $M$, we train a Small PT for 2M steps at batch size 512 (rebuttal Fig. 1A and Fig. 2A right panel). Note, this is 8x as many examples per task as our base setting (500K steps at batch size 256). Despite this, the location of the task diversity threshold does not change. If the task diversity threshold were to disappear in the ideal experiment described above, then in the experiment we presented, we would expect the threshold to increase at least by a factor of 2 when we increase training data from 2x to 8x examples per task. But this is not the case, suggesting that the location of the threshold is robust in this setting.
2. Of course, this does not eliminate the possibility that $t^*>$ 2M: with finite training time experiments, we cannot prove that models after the task diversity threshold will not eventually converge to dMMSE. However, in rebuttal Fig. 2B, we study how the early stopping time for Ridge, $t^*$, scales with $M$. For most $M<M^*$, $t^*$ obeys a simple scaling behavior $t^* \propto M^\alpha$ with $\alpha \approx 0.47$. However, for $M>2^{10}$, the distance to Ridge decreases monotonically through training and $t^*=$2M. While it is possible that, at some $t>$ 2M steps, the model’s distance to Ridge will start increasing again, the $t^*$ would be much larger than what is predicted by the scaling relationship. The break in the scaling behavior near the task diversity threshold suggests that our findings are not just caused by undertraining. There is a difference in the training dynamics and the dMMSE solution, if not unreachable, at least takes an uncharacteristically long time to find.
3. In Fig 2A, we visualize the training curves for the models at batch size 512 for both 500K and 2M steps for $M$ before and after the threshold as well as the training curve for the model trained with infinite task diversity. For both the shorter and longer training durations, models trained with the same task diversity have similar qualitative behavior (whether distance to Ridge decreases then increases or monotonically decreases). Additionally, the learning curves of the models beyond the task diversity threshold are very similar to the learning curves for models trained on infinite pretraining task diversities (Fig 2A) and they achieve similar final accuracies (dashed lines vs markers in Fig 1A), suggesting that these models really are approaching the ridge solution, and will not depart from it even if trained longer.

**A possible picture/explanation for the task diversity threshold.** Multiple reviewers have asked why there is a task diversity threshold phase transition. While we don’t like to speculate, at the reviewers’ request, we provide our best current guess given the new rebuttal experiments. Indeed the result in Fig. 2 suggests a natural loss landscape explanation for the task diversity threshold phase transition. The pre-training loss landscape always has a local minimum corresponding to dMMSE. We conjecture: at low task diversity ($M < 2^{11}$), this local minimum is wide enough to eventually attract the training trajectory. We further conjecture: as task diversity increases, the local minimum corresponding to dMMSE becomes sharper/its basin of attraction becomes smaller until it can no longer attract training trajectories. After the task diversity threshold, models do not find the sharp dMMSE solution no matter how long you train, but instead, go to a higher training error but wider minimum that corresponds to Ridge. While this is speculation, it does suggest many interesting loss landscape analyses that would be exciting follow up works to our work. In this sense, we hope our discovery and substantiation of a task diversity threshold phase transition constitutes an important and impactful contribution to the NeurIPS community and inspires follow up work.

---

### Decision · Program_Chairs · 2023-09-21

**Decision:**

Accept (poster)

**Comment:**

The submission contributes hypothesis-driven experiments to a highly-discussed subarea (in-context learning / few-shot generalization in transformers). Though several reviewers noted the gap between the setting examined in the submission and the broader settings in which transformers are employed, all reviewers found the submission thought-provoking with the potential to spur more conclusive investigations into why the factor identified in the work (data diversity in pre-training) matters.